# Direct integration of non-axisymmetric Gaussian wind-turbine wake including yaw and wind-veer effects

**Karim Ali, Pablo Ouro, and Tim Stallard**

School of Engineering, University of Manchester, Manchester, M13 9PL, UK

**Correspondence:** Pablo Ouro (pablo.ouro@manchester.ac.uk)

**Abstract.** TS1 The performance of a wind farm is significantly influenced by turbine–wake interactions. These interactions are typically quantified for each turbine either by measuring its nacelle wind speed or by evaluating its rotor-averaged wind speed using numerical methods that involve a set of discrete points across the rotor disc. Although various point distributions exist in the literature, we introduce two analytical expressions for integrating non-axisymmetric Gaussian wakes, which account for wake stretching and shearing resulting from upstream turbine yaw and wind veer. The analytical solutions correspond to modelling the target turbine as a circular actuator disc and as an equivalent rectangular actuator disc. The derived expressions are versatile, accommodating any offset and hub-height difference between the wake source (upstream turbine) and the target turbine. Verification against numerical evaluations of the rotor-averaged deficit using 2000 averaging points at various downstream locations from the wake source demonstrates excellent agreement for both analytical solutions at small/moderate veer effects, whereas only the equivalent rectangular-disc solution was accurate under extreme veer conditions. In terms of computational cost compared to vectorised numerical averaging using 16 averaging points, both analytical solutions are computationally efficient with the circular-disc solution being approximately 15 % slower and the rectangular-disc solution being approximately 10 % faster. Furthermore, the analytical expressions are shown to be compatible with multiple wake superposition models and are differentiable, providing a foundation for deriving analytical gradients which can be advantageous for optimisation-based applications.

## 1 Introduction

The widespread deployment of wind farms necessitates the use of accurate and efficient computational tools for preliminary design and optimisation (Veers et al., 2023). While computational fluid dynamic (CFD) methods such as large-eddy simulation (LES) and Reynolds-averaged Navier–Stokes (RANS) offer detailed insights into turbine loading and wake dynamics, they are often too computationally intensive for preliminary wind-farm design and layout optimisation (Maas and Raasch, 2022). Mesoscale models require less computational power and have been employed to examine the large-scale interactions between wind-turbine wakes and the turbulent atmospheric boundary layer, though they lack the detailed wake resolution provided by RANS and LES models (Fitch et al., 2012; Ali et al., 2023). Conversely, engineering wake models are comparatively faster

and are extensively used in various wind-energy applications, including wind-farm layout optimisation and control (Hou et al., 2016; Bay et al., 2018; Shapiro et al., 2022). Engineering wake models, which assume that a turbine's wake is self-similar, represent the wake using a streamwise scaling deficit function and a shape function to describe the deficit distribution perpendicular to the streamwise direction. Various shape functions have been proposed, including top-hat profiles (Jensen, 1983), Gaussian profiles (Bastankhah and Porté-Agel, 2014), double-Gaussian profiles (Keane et al., 2016), super-Gaussian profiles (Blondel and Cathelain, 2020; Ouro and Lazennec, 2021), cosine-bell profiles (Jensen, 1983; Zhang et al., 2020), and profiles based on scalar diffusion (Cheng and Porté-Agel, 2018; Ali et al., 2024d). Among these, the Gaussian wake TS2 profile

is widely adopted particularly for distances comparable to a typical inter-turbine spacing within a wind farm.

To assess the impact of an upstream turbine's wake on the onset flow of a downstream rotor, such as required to estimate the reduction of available kinetic energy flux due to wake effects, numerical methods often average the upstream deficit calculated at multiple control points across the rotor disc of the considered turbine. The number and distribution of these averaging points vary in the literature. Allaerts and Meyers (2019) employed a 16-point quadrature based on Holoborodko (2011) in their analysis of wind-farm blockage and induced gravity waves, whereas Stipa et al. (2024) utilised a cross-like distribution of 16 averaging points to enhance radial resolution across the rotor (see Fig. E2). Stanley and Ning (2019) used 100 equally spread averaging points for the evaluation of the rotor-averaged deficit. Other studies proposed uniform radial and azimuthal distribution of averaging points across the rotor within the context of farm layout optimisation and control (Li et al., 2022; Ling et al., 2024).

Uncertainties can arise from the number, distribution, and averaging weights of the control points, especially when the shape of the upstream wake deviates from the axisymmetric form due to, for instance, wind-veer effects. Rather than numerical averaging, Ali et al. (2024a) developed an analytical expression for the circular-disc integration of an axisymmetric Gaussian function depicting the wake of an upstream turbine. Their formulation is applicable to any offset between the upstream turbine (wake source) and the considered turbine, but it assumes that the upstream wake is axisymmetric and that both the upstream and downstream turbines have the same hub height. Typically, turbines can be yawed relative to their onset wind, yielding wakes that are not axisymmetric but rather of elliptic shape (Bastankhah and Porté-Agel, 2016). Additionally, wind-veer effects can cause planar shearing of the wake shape through stretching the wake elliptic contours and rotating their major axes (see Fig. 1 later in the article), resulting in further deviation from the axisymmetric wake shape (Abkar et al., 2018). Furthermore, onshore wind farms often have turbines with different hub heights due to non-uniform terrain, and offshore wind farms may have turbines of varying hub heights and diameters operating in close proximity.

In this study, we extend the analytical solution proposed by Ali et al. (2024a) by generalising the assumed upstream wake shape to include non-symmetry due to yawing of the wake source, wind-veer effects, and different hub heights between the wake source and target turbine. The primary focus is on wind-turbine wakes, but the proposed expressions are also applicable to tidal-stream turbines and can be extended to vertical-axis turbines (both wind and tidal) due to the relevance of similar Gaussian wake profiles (e.g. Stallard et al., 2015; Ouro and Lazennec, 2021). Although rotor-induction effects can alter the onset wind profile of the considered turbine, we do not consider these effects similar to various engineering wake models. Additionally, we assume that the considered turbine is modelled as a uniform actuator disc, corresponding to uniform averaging weights across the turbine's rotor, and that the effects of blade geometry are neglected. The objective of the proposed analytical solutions is not to replace numerical approaches, which are the only available option for arbitrary wind-speed fields, but to provide an alternative approach in the specific case of Gaussian wakes. Furthermore, analytical solutions can be computationally cheaper than numerical approaches and for some scenarios (such as high wind veer) can be more accurate than numerical averaging at common resolutions from the literature.

The surface integration of a Gaussian field across a circular disc is often complicated because of the modified Bessel function that arises from the azimuthal integration of a shifted Gaussian function. As will be discussed later, the analytical solution of the rotor-averaged deficit over a circular disc will be derived based on some simplifying assumptions that limit the validity range of the analytical solution (more details in Sect. 2.2). Conversely, the surface integration of a Gaussian field across an equivalent rectangular disc often has a closed-form analytical solution without the need to limit the solution's validity. By appropriate sizing of the rectangular disc of integration, highly accurate approximate analytical solutions of the surface integration across a circular disc can be obtained. DiDonato and Jarnagin (1961) used the circle–rectangle analogy to approximate the circular-disc integration of an elliptic Gaussian field using look-up tables of the error function. Furthermore, Ali et al. (2024d) obtained an approximate analytical solution of a complicated two-dimensional integration involving the modified Bessel function by making use of the circle–rectangle analogy based on the analytical solution of Ali et al. (2024b). Cheung et al. (2024) used the same analogy to obtain analytical solutions of a turbine's induction effects under various conditions using a Green-function approach. As such, we also derive an analytical solution of the rotor-averaged deficit for an equivalent rectangular disc, which is not limited by the simplifying assumptions of the circular-disc integration.

The rest of this paper is structured as follows. Section 2 presents generalised analytical expressions for the rotor-averaged deficit in the case of a circular disc (Sect. 2.2) and an equivalent rectangular disc (Sect. 2.3), which are verified against numerical solutions for a single upstream wake (Sect. 3.1) and for multiple upstream wakes (Sect. 3.4). The effect of some relevant parameters on the rotor-averaged deficit is discussed in Sect. 3.2 and 3.3. The computational costs of the proposed solutions compared to numerical approaches are examined in Sect. 3.5, and their accuracy against various numerical resolutions is quantified in Sect. 3.6. The key findings of this paper are discussed in more detail in Sect. 4 with a focus on compatibility with different wake superposition models, with a summary in Sect. 5. Appendices A–D contain mathematical details on the derivation of the generalised rotor-averaged deficit, whereas vari-

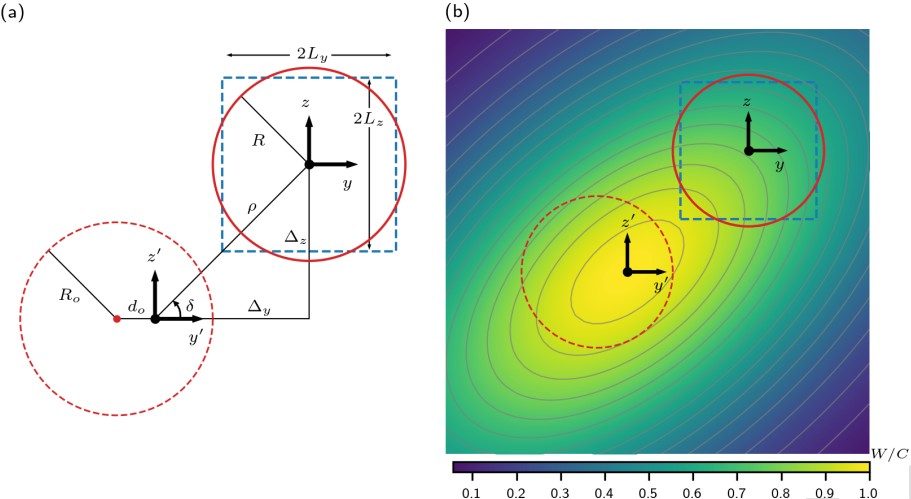

**Figure 1.** **(a)** Schematic of the wake axes ($y'$–$z'$) and the axes of the considered turbine ($y$–$z$) separated by the distances ($\Delta_y$, $\Delta_z$) with polar coordinates $\rho$ and $\delta$. The upstream turbine (wake source) is represented by the dashed red circle with radius $R_o$, whereas the considered turbine is represented by the solid red circle with radius $R$. The wake centre is deflected horizontally by $d_o$ from the centre of the upstream turbine (red dot). An equivalent rectangle of the considered turbine (Sect. 2.3) is shown in dashed blue with dimensions of $2L_y$ and $2L_z$ in $y$ and $z$ directions, respectively. **(b)** Sample contours of the normalised wind-speed deficit $W/C$ (Eq. 1) calculated at an eccentricity $\xi = 0.4$ and a veer coefficient $\omega = -0.6$, where the definitions of $\xi$ (Eq. 3) and $\omega$ (Eq. 2) are provided in the main text. The red circles, the blue rectangle, and the axes $y'$–$z'$ and $y$–$z$ have the same definitions as in **(a)**.

ous resolutions and distributions of averaging points are summarised in Appendix E. Further mathematical manipulations regarding wake superposition are included in Appendix F following the discussion in Sect. 4. Some additional material is included in Appendix G.

## 2 Generalised rotor-averaged deficit of an elliptic veered Gaussian wake

In this study, we seek to analytically evaluate a rotor-averaged deficit of a turbine operating within an upstream Gaussian wake whose shape and centre are defined. For simplicity, the expression for the rotor-averaged deficit is derived for a single upstream wake, but extension to multiple upstream wakes is straightforward (Sect. 3.4). Some key definitions are presented first in Sect. 2.1, followed by deriving analytical solutions in the case of a circular disc (Sect. 2.2) and an equivalent rectangular disc (Sect. 2.3). The presented analysis is applicable to any engineering wake model that utilises the Gaussian wake profile to describe the wake shape normal to the streamwise direction.

### 2.1 Problem definition

The normalised wind-speed deficit ($W$) due to the wake of an upstream turbine impacted by a constant transverse wind (causing wind veer) can be expressed as (Bastankhah and

Porté-Agel, 2016; Abkar et al., 2018)

$$
\begin{aligned}
W(x, y', z') &= 1 - \frac{u(x, y', z')}{\overline{u}_o} \\
&= C(x) \exp\left(\frac{-(y' + \omega z')^2}{2\sigma_y^2}\right) \exp\left(\frac{-z'^2}{2\sigma_z^2}\right),
\end{aligned} \quad (1)
$$

where $u$ is the streamwise wind speed, $\overline{u}_o$ is the rotor-averaged wind speed of the upstream turbine (wake source), $C$ is a streamwise scaling function, and $x$ is the streamwise distance between the two turbines. The variables $y'$ and $z'$ are the lateral and vertical coordinates, respectively, in a plane normal to the streamwise direction with an origin at the wake centre, and

$$
\omega = \Delta\alpha_o\left(\frac{x}{D_o}\right) \quad (2)
$$

is a wind-veer coefficient with $\Delta\alpha_o$ being the difference in wind direction across the top and bottom tips of the upstream turbine (wake source) whose diameter is $D_o$. The quantities $\sigma_y$ and $\sigma_z$ are the wake standard deviations in $y'$ and $z'$ directions, respectively. Figure 1 illustrates a schematic of an upstream turbine (of radius $R_o$) whose wake centre is deflected horizontally by a distance $d_o$. The Cartesian axes $y'$–$z'$ are placed at the centre of the wake in the plane containing the considered turbine which is at a streamwise distance $x$ from the wake source. The centre of the considered turbine (of radius $R$) is located at ($\Delta_y$, $\Delta_z$) with respect to the wake centre with polar coordinates $\rho$ and $\delta$. The Cartesian axes $y$–$z$ are

placed at the centre of the considered turbine. The offset $\rho$ is measured from the centre of the wake, which is assumed to be known from wake deflection models (e.g. Bastankhah and Porté-Agel, 2016; Qian and Ishihara, 2018; Snaiki and Makki, 2024).

For a yawed upstream turbine, the wake standard deviations $\sigma_y$ and $\sigma_z$ are not equal, resulting in elliptic wake contours rather than circular contours in the specific case of axisymmetric wake. We define the eccentricity $\xi \geq 0$ of the wake elliptic contours due to having non-equal $\sigma_y$ and $\sigma_z$ as

$$\xi = \sqrt{1 - \left(\frac{\sigma_y}{\sigma_z}\right)^2}. \tag{3}$$

Here, it is assumed that $\sigma_y \leq \sigma_z$, which is the typical case for yawed horizontal-axis wind turbines. However, it is noted that where relevant, scenarios with $\sigma_y > \sigma_z$ can be obtained by a rotation of axes. In the following calculations, $\sigma_z$ will be denoted as $\sigma$, and hence $\sigma_y = \sigma\sqrt{1 - \xi^2}$. A typical range for the eccentricity $\xi$ can be identified using the empirical expressions for $\sigma_y$ and $\sigma_z$ for a yawed upstream turbine at a yaw misalignment $\gamma_o$:

$$\sigma_z = \sigma = k_z^* x + \sigma_{z_0} D_o,$$

and $\sigma_y = \sigma\sqrt{1 - \xi^2} = k_y^* x + \sigma_{z_0} D_o \cos\gamma_o, \tag{4}$

where $k_z^*$ and $k_y^*$ are the rates of wake expansion in $z'$ and $y'$ directions, respectively, and $\sigma_{z_0} \approx 1/\sqrt{8}$ is an initial wake standard deviation (Bastankhah and Porté-Agel, 2016). For simplicity we assume that $k_z^* \approx k_y^* = k^*$, and hence

$$\xi^2 \approx 1 - \left(\frac{k^* x/\sigma_{z_0} + \cos\gamma_o}{k^* x/\sigma_{z_0} + 1}\right)^2 \leq 1 - \cos^2\gamma_o. \tag{5}$$

The typical range of a turbine yaw angle is less than 30° (Zong and Porté-Agel, 2021), and hence the eccentricity of the wake elliptic contours is $\xi < 1/2$ for a typical inter-turbine spacing.

A Gaussian wake description, as given in Eq. (1), assumes a neutral atmospheric boundary layer for which the typical magnitude of wind veer is approximately of the order of 0.03° m$^{-1}$ (Walter et al., 2009; Gao et al., 2021). Hence, for a large wind turbine (diameter $\sim 220$ m) operating in a neutral boundary layer, the difference in wind direction across its top and bottom tips is less than approximately 7°. While stable stratification and/or complex terrain can intensify wind veer (Ghobrial et al., 2024), we limit our calculations for the case of a circular disc (Sect. 2.2) to neutral boundary layers with moderate wind veer (i.e. $\Delta\alpha_o \lesssim 7°$). The expression derived for the equivalent rectangular disc (Sect. 2.3) will not be limited by the moderate-veer assumption.

The angle $\delta$ corresponds to the difference in hub height between the upstream turbine (wake source) and the considered downstream turbine. In a typical wind farm, all turbines

have the same hub height, making $\delta = 0$ (or $\pi$). However, our calculations consider $\delta$ as a variable to accommodate cases with differing hub heights, such as adjacent wind farms or non-uniform terrain. Rather than using linear averaging of the wind-speed deficit across the disc of integration, we generalise the averaging process to an order of $n > 0$ such that

$$\overline{W}^{(n)} = \left(\frac{1}{A}\iint\limits_A W^n \, dA\right)^{1/n}, \tag{6}$$

where $\overline{W}^{(n)}$ is the $n$th-order rotor-averaged deficit, $n$ is the averaging order, and $A$ is the area of the disc depicting the turbine (circular in Sect. 2.2 and rectangular in Sect. 2.3). As such, if $n = 2$, then a root-mean-square averaging of the deficit across the rotor is obtained.

To summarise, the objective is to determine the rotor-averaged deficit of a turbine of radius $R$ operating within an upstream Gaussian wake defined by the standard deviation $\sigma$, the wake eccentricity $\xi$, the veer coefficient $\omega$, and the streamwise scaling function $C$, by performing the surface integration in Eq. (6) following the definition of the normalised deficit $W$ in Eq. (1). The considered turbine is offset from the wake centre by the radial distance $\rho$ and the angle $\delta$. We assume that the rotor disc of the considered turbine is normal to the free-stream direction (non-yawed rotor), implying that $\sigma$, $\xi$, $\omega$, and $C$ are variables in the streamwise direction only. If, however, the considered turbine is yawed, this simplifying assumption has no significant impact on the rotor-averaged deficit for small yaw angles (i.e. $\gamma \lesssim 30°$), as well established in the literature. Specifically, the relative error of the rotor-averaged deficit from this simplifying assumption is of the order of $k^{*2}\sin^2\gamma \sim \mathcal{O}(10^{-3})$, which is negligible. Additionally, a yawed turbine can experience transverse wind whose magnitude is typically much smaller than the streamwise wind speed (Martínez-Tossas et al., 2019). This transverse wind is not included in the following analysis and needs to be modelled numerically, if required. However, the streamwise wind speed is dominant for small yaw angles, making the following analysis applicable to turbines of small yaw angles with no significant loss of accuracy.

## 2.2 Analytical rotor-averaged deficit across a circular disc

The derivation in this section is a generalisation to the solution by Ali et al. (2024a), who solved a linear version of the problem (i.e. $n = 1$) but for an axisymmetric wake (i.e. $\xi = \omega = 0$) and for two turbines of the same hub height (i.e. $\delta = 0$). For a circular disc of radius $R$ and by using the definition of $W$ (Eq. 1), Eq. (6) becomes TS3

$$\frac{\overline{W}_c^{(n)}}{C} = \left(\frac{1}{\pi R^2}\int\limits_0^R\int\limits_0^{2\pi} r \exp\left(\frac{-n(y' + \omega z')^2}{2\sigma_y^2}\right)\exp\left(\frac{-nz'^2}{2\sigma_z^2}\right) d\theta \, dr\right)^{1/n}, \tag{7}$$

where $\overline{W}_{\mathrm{c}}^{(n)}$ is the rotor-averaged deficit of the order $n$ across a circular disc, and $r$ and $\theta$ are the polar coordinates of the $y$–$z$ axes placed at the centre of the considered turbine (Fig. 1). The coordinates $y'$–$z'$ (of the wake centre) and $y$–$z$ can be related using $y' = y + \Delta_y$ and $z' = z + \Delta_z$ (Fig. 1). These relations, along with $\langle y, z \rangle = r \langle \cos\theta, \sin\theta \rangle$ and $\langle \Delta_y, \Delta_z \rangle = \rho \langle \cos\delta, \sin\delta \rangle$, where $\langle t_1, t_2 \rangle$ means $t_1$ or $t_2$, can be used to re-write Eq. (7) in the $r$–$\theta$ coordinates as (see Appendix A for derivation)

$$
\frac{\overline{W}_{\mathrm{c}}^{(n)}}{C} = \exp\left(\frac{-\rho^2}{2\sigma_*^2}\right) \exp\left(\frac{-\rho^2 \cos(2\delta - \phi_{\mathrm{ns}})}{2\sigma_{\mathrm{ns}}^2}\right)
$$
$$
\left(\underbrace{\int_0^1 \eta \exp\left(\frac{-n\eta^2 R^2}{2\sigma_*^2}\right) M_\theta \, \mathrm{d}\eta}_{M_\eta}\right)^{1/n}, \tag{8}
$$

where $\eta = r/R$, and the integral $M_\theta$ is

$$
M_\theta = \frac{1}{\pi} \int_0^{2\pi} \exp\left(\frac{-n\eta^2 R^2 \cos(2\theta - \phi_{\mathrm{ns}})}{2\sigma_{\mathrm{ns}}^2}\right)
$$
$$
\exp\left(\frac{-n\eta R\rho \cos(\theta - \phi_{\mathrm{s}})}{\sigma_{\mathrm{s}}^2}\right) \mathrm{d}\theta. \tag{9}
$$

In Eqs. (8) and (9), three new length scales are introduced: $\sigma_*$, $\sigma_{\mathrm{ns}}$, and $\sigma_{\mathrm{s}}$. In addition, there are two new angles: $\phi_{\mathrm{ns}}$ and $\phi_{\mathrm{s}}$, which are defined in terms of the wake standard deviation $\sigma$, the eccentricity $\xi$, the veer coefficient $\omega$, and the angle $\delta$ as

$$
\frac{\sigma_*^2}{\sigma^2} = \frac{2(1-\xi^2)}{2+\omega^2-\xi^2}, \quad \frac{\sigma_{\mathrm{ns}}^2}{\sigma^2} = \frac{2(1-\xi^2)}{\sqrt{(\omega^2-\xi^2)^2+4\omega^2}},
$$
$$
\frac{\sigma_{\mathrm{s}}^2}{\sigma^2} = \frac{(1-\xi^2)\cos\phi_{\mathrm{s}}}{\cos\delta + \omega\sin\delta}, \quad \tan\phi_{\mathrm{ns}} = \frac{2\omega}{\xi^2-\omega^2},
$$
$$
\tan\phi_{\mathrm{s}} = \omega + \frac{(1-\xi^2)\tan\delta}{1+\omega\tan\delta}. \tag{10}
$$

The subscript "ns" refers to wake non-symmetry. In the case of an axisymmetric wake (i.e. $\omega = \xi = 0$), we have $\sigma_{\mathrm{ns}}^{-1} = 0$, and hence its corresponding exponential terms in Eqs. (8) and (9) vanish. Also, when the wake is axisymmetric we have $\sigma_* = \sigma_{\mathrm{s}} = \sigma$ and $\phi_{\mathrm{s}} = \delta$. The solution to the integral $M_\theta$ in Eq. (9) is (see derivation in Appendix B)

$$
M_\theta = 2I_0\left(\frac{n\eta R\rho}{\sigma_{\mathrm{s}}^2}\right) I_0\left(\frac{n\eta^2 R^2}{2\sigma_{\mathrm{ns}}^2}\right)
$$
$$
+ 4\sum_{\nu \geq 1} (-1)^\nu \cos(\nu\phi) \, I_{2\nu}\left(\frac{n\eta R\rho}{\sigma_{\mathrm{s}}^2}\right) I_\nu\left(\frac{n\eta^2 R^2}{2\sigma_{\mathrm{ns}}^2}\right), \tag{11}
$$

where $I_\nu$ is the modified Bessel function of the first kind and integer order $\nu$, and $\phi = 2\phi_{\mathrm{s}} - \phi_{\mathrm{ns}}$. By employing Eq. (11),

an approximate solution of the integral $M_\eta$ is (Appendix C)

$$
M_\eta \approx 2\mu_0^{(n)}\left(1 + 2\mathcal{P}_{\mathrm{ns}}^{(n)}\right) - \frac{4\sigma_*^2 \mathcal{P}_{\mathrm{ns}}^{(n)}}{nR^2} \exp\left(\frac{-nR^2}{2\sigma_*^2}\right)
$$
$$
\left[\frac{\lambda}{\rho} I_1\left(\frac{nR\rho}{\sigma_{\mathrm{s}}^2}\right) + \frac{\lambda^2}{\rho^2} I_2\left(\frac{nR\rho}{\sigma_{\mathrm{s}}^2}\right)\right], \tag{12}
$$

where TS4 $\quad \lambda = R\sigma_{\mathrm{s}}^2/\sigma_*^2$, and $\quad \mathcal{P}_{\mathrm{ns}}^{(n)} = \cos(n\chi_{\mathrm{ns}}^2 \sin\phi)$ $\exp\left(-n\chi_{\mathrm{ns}}^2 \cos\phi\right) - 1$ with $\chi_{\mathrm{ns}} = \rho\sigma_*^2/(2\sigma_{\mathrm{ns}}\sigma_{\mathrm{s}}^2)$. In Eq. (12), $\mu_0^{(n)}$ is

$$
\mu_0^{(n)} = \int_0^1 \eta \exp\left(\frac{-n\eta^2 R^2}{2\sigma_*^2}\right) I_0\left(\frac{n\eta R\rho}{\sigma_{\mathrm{s}}^2}\right) \mathrm{d}\eta. \tag{13}
$$

In the case of an axisymmetric wake ($\sigma_{\mathrm{ns}}^{-1} = 0$), we have $\chi_{\mathrm{ns}} = \mathcal{P}_{\mathrm{ns}}^{(n)} = 0$, and Eq. (12) simplifies to $M_\eta \approx 2\mu_0^{(n)}$. Therefore, Eq. (12) indicates that the solution of the non-axisymmetric wake (Eq. 1) is a perturbation (second term in Eq. (12) TS5 to a scaled axisymmetric solution (scaled by $1 + 2\mathcal{P}_{\mathrm{ns}}^{(n)}$). Additionally, Eq. 12 contains terms in the form $I_\nu(nR\rho/\sigma_{\mathrm{s}}^2)/\rho$, which has a finite value when there is no offset between the wake source and the considered turbine ($\rho = 0$) as TS6 $\lim_{\rho \to 0} I_\nu(nR\rho/\sigma_{\mathrm{s}}^2)/\rho = 1/(2^\nu \nu!)$. Nonetheless, at no offset ($\rho = 0$), we have $\mathcal{P}_{\mathrm{ns}}^{(n)} = 0$, similar to the axisymmetric solution. This results from the simplifying assumption made in Appendix C to solve for $M_\eta$, where the terms $I_\nu\left(n\eta^2 R^2/(2\sigma_{\mathrm{ns}}^2)\right)$ were approximated by $(n\eta^2 R^2/(4\sigma_{\mathrm{ns}}^2))^\nu/\nu!$ under the assumption that the argument of the modified Bessel function is small (following the limits on wind veer discussed in Sect. 2.1), and hence $I_0\left(n\eta^2 R^2/(2\sigma_{\mathrm{ns}}^2)\right) \sim 1$ was employed. This means that the stretching and shearing acting on the wake are assumed to have minimal effect on the wake shape close to the wake centre and are more profound far from the wake centre. We will show in Sect. 3.1 that this assumption is acceptable for moderate values of wind veer by monitoring the average value (within the range $0 \leq \eta \leq 1$) of the argument of the modified Bessel function $\kappa^{(n)}$, defined as

$$
\kappa^{(n)} = \frac{nR^2}{2\sigma_{\mathrm{ns}}^2} \int_0^1 \eta^2 \, d\eta = \frac{nR^2}{6\sigma_{\mathrm{ns}}^2}. \tag{14}
$$

The parameter $\kappa^{(n)}$ is a measure of the skewness of the wind-speed deficit within the rotor of the considered turbine. When the wake is axisymmetric (i.e. no skewness), we have $\kappa^{(n)} = 0$. As the shearing and stretching of the upstream wake contours increase, the value of $\kappa^{(n)}$ increases, which can also be raised by the averaging order $n$. In Sect. 3.2 and 3.3, it will be shown that a practical limit on $\kappa^{(n)}$ for the circular-disc solution is around 0.4–0.5, and higher values could result in larger deviation from the numerical solution.

The solution of the integral $\mu_0^{(n)}$ can be obtained by generalising the solution introduced by Ali et al. (2024a) based on Rosenheinrich (2017):

$$\mu_0^{(n)} = \frac{\sigma_*^2}{nR^2} \exp\left(\frac{-nR^2}{2\sigma_*^2}\right) \Psi^{(n)}(R, \rho, \sigma_s, \sigma_*), \quad (15)$$

where

$$\Psi^{(n)}(R, \rho, \sigma_s, \sigma_*) = I_0\left(\frac{nR\rho}{\sigma_s^2}\right) \sum_{k\geq 1}\left[\left(\frac{nR^2}{2\sigma_*^2}\right)^k f_k(n\tau^2)\right]$$
$$- \frac{nR\rho}{\sigma_s^2} I_1\left(\frac{nR\rho}{\sigma_s^2}\right) \sum_{k\geq 1}\left[\left(\frac{nR^2}{2\sigma_*^2}\right)^k g_k(n\tau^2)\right], \quad (16)$$

and $\tau = \rho\sigma_*/\sigma_s^2$. The coefficients $f_k$ and $g_k$ follow the recursions

$$f_k(v) = \frac{f_{k-1}(v) + v g_{k-1}(v)}{k}, \quad g_k(v) = \frac{f_k(v) + 2g_{k-1}(v)}{2k}, \quad (17)$$

with $f_0 = 1$ and $g_0 = 0$. The recursions in Eq. (17) converge rapidly within 6–10 iterations of simple algebraic calculations (scalar addition and multiplication). From Eq. (8), the final form of the rotor-averaged deficit is

$$\overline{W}_c^{(n)}/C \approx \exp\left(\frac{-\rho^2}{2\hat{\sigma}^2}\right)\left(2\mu_0^{(n)}\left(1 + 2\mathcal{P}_{ns}^{(n)}\right)\right.$$
$$- \frac{4\sigma_*^2 \mathcal{P}_{ns}^{(n)}}{nR^2}\exp\left(\frac{-nR^2}{2\sigma_*^2}\right)\left[\frac{\lambda}{\rho}I_1\left(\frac{nR\rho}{\sigma_s^2}\right)\right.$$
$$\left.\left. + \frac{\lambda^2}{\rho^2}I_2\left(\frac{nR\rho}{\sigma_s^2}\right)\right]\right)^{1/n}, \quad (18)$$

where $\hat{\sigma}^{-2} = \sigma_*^{-2} + \sigma_{ns}^{-2}\cos(2\delta - \phi_{ns})$. Equation (18) was implemented in Python and is available from Ali et al. (2024c).

## 2.3 Analytical rotor-averaged deficit across a rectangular disc

As discussed in Sect. 2.2, the derived expression for the rotor-averaged deficit, assuming a circular-disc representation of the considered turbine (Eq. 18), is valid when the skewness parameter $\kappa^{(n)}$ is small (Eq. 14; approximately less than 0.4–0.5). However, when $\kappa^{(n)}$ is large because of strong veer and/or large averaging order $n$, Eq. (18) might no longer be valid or become of poor accuracy. As such, we derive herein an alternative expression for the rotor-averaged deficit assuming a rectangular-disc representation of the considered turbine following similar analogies in the literature (Ali et al., 2024d; Cheung et al., 2024). The dimensions of the rectangular disc are $2L_y$ and $2L_z$ in $y$ and $z$ directions, respectively, with the same centre as the considered turbine (Fig. 1). We start by re-writing Eq. (6) for a rectangular disc with the aid of the definition of $W$ in Eq. (1) as

$$\overline{W}_r^{(n)}/C = \left(\frac{1}{4L_yL_z}\int_{\Delta_z - L_z}^{\Delta_z + L_z} dz' \exp\left(\frac{-nz'^2}{2\sigma_z^2}\right)\right.$$
$$\left.\int_{\Delta_y - L_y}^{\Delta_y + L_y} dy' \exp\left(\frac{-n(y' + \omega z')^2}{2\sigma_y^2}\right)\right)^{1/n}, \quad (19)$$

where $\overline{W}_r^{(n)}$ is the $n$th-order rotor-averaged deficit for a rectangular disc. The solution of the inner integral (over $y'$) is [TS7]

$$\int_{\Delta_y - L_y}^{\Delta_y + L_y} \exp\left(\frac{-n(y' + \omega z')^2}{2\sigma_y^2}\right) dy'$$
$$= \sqrt{\frac{\pi(1-\xi^2)}{2n}}\sigma\left(\text{erf}\left(\frac{\Delta_y + L_y + \omega z'}{\sigma\sqrt{2(1-\xi^2)/n}}\right)\right.$$
$$\left. - \text{erf}\left(\frac{\Delta_y - L_y + \omega z'}{\sigma\sqrt{2(1-\xi^2)/n}}\right)\right), \quad (20)$$

[TS8] where $\sigma_y = \sigma\sqrt{1 - \xi^2}$ (Eq. 4), and erf is the error function defined as (Ng and Geller, 1969, 3.1; 1)

$$\text{erf}(h) = \frac{2}{\sqrt{\pi}}\int_0^h \exp\left(-s^2\right) ds. \quad (21)$$

As such, Eq. (19) becomes

$$\overline{W}_r^{(n)}/C = \left(\frac{\sigma}{4L_yL_z}\sqrt{\frac{\pi(1-\xi^2)}{2n}}(Q_1 - Q_2)\right)^{1/n}, \quad (22)$$

where $Q_1$ and $Q_2$ are defined as

$$Q_{\langle 1,2\rangle} = \int_{\Delta_z - L_z}^{\Delta_z + L_z} \exp\left(\frac{-nz'^2}{2\sigma^2}\right)\text{erf}\left(\frac{\Delta_y \pm L_y + \omega z'}{\sigma\sqrt{2(1-\xi^2)/n}}\right) dz', \quad (23)$$

and the $\pm$ sign in Eq. (23) corresponds to $Q_1$ and $Q_2$, respectively. We can solve for the integrals $Q_1$ and $Q_2$ by making use of the generalised Owen's T function $\Omega(h, a, b)$ defined as (Przemo, 2019)

$$\Omega(h, a, b) = \frac{1}{2\sqrt{2\pi}}\int_h^\infty \exp\left(\frac{-s^2}{2}\right)\text{erf}\left(\frac{as+b}{\sqrt{2}}\right) ds$$
$$= \frac{1}{2\pi}\left(\underbrace{\arctan(a)} - \arctan(a + b/h)\right.$$
$$\left. - \arctan\left(\frac{h + ab + a^2h}{b}\right)\right) + \underbrace{\frac{1}{4}\text{erf}\left(\frac{b}{\sqrt{2(1-a^2)}}\right)}$$
$$+ \text{T}(h, a + b/h) + \text{T}\left(\frac{b}{\sqrt{1+a^2}}, \frac{h + ab + a^2h}{b}\right), \quad (24)$$

where $T(h, a)$ is Owen's T function defined as (Owen, 1956)

$$T(h, a) = \frac{1}{2\pi} \int_0^a \frac{1}{1+s^2} \exp\left(\frac{-h^2(1+s^2)}{2}\right) \, ds. \tag{25}$$

From the definition of the function $\Omega$ (Eq. 24) along with $\sigma_z = \sigma$ (Eq. 4), we can express the integrals $Q_1$ and $Q_2$ as

$$Q_{\langle 1,2 \rangle} = 2\sqrt{\frac{2\pi}{n}}\sigma\left[\Omega\left(\frac{\Delta_z - L_z}{\sigma/\sqrt{n}}, \frac{\omega}{\sqrt{1-\xi^2}}, \frac{\Delta_y \pm L_y}{\sigma\sqrt{(1-\xi^2)/n}}\right) \right.$$
$$\left. - \Omega\left(\frac{\Delta_z + L_z}{\sigma/\sqrt{n}}, \frac{\omega}{\sqrt{1-\xi^2}}, \frac{\Delta_y \pm L_y}{\sigma\sqrt{(1-\xi^2)/n}}\right)\right]. \tag{26}$$

Combining Eqs. (26) and (22) gives the disc-averaged deficit for a rectangular disc as

$$\frac{\overline{W}_r^{(n)}}{C} = \left(\frac{\pi\sigma^2\sqrt{1-\xi^2}}{2nL_yL_z}\sum_{s_y, s_z \in \{-1,1\}} (-s_ys_z)\right.$$
$$\left.\Omega\left(\frac{\Delta_z + s_zL_z}{\sigma/\sqrt{n}}, \frac{\omega}{\sqrt{1-\xi^2}}, \frac{\Delta_y + s_yL_y}{\sigma\sqrt{(1-\xi^2)/n}}\right)\right)^{1/n}. \tag{27}$$

The expression in Eq. (27) simply calculates the function $\Omega$ (Eq. 24) at the four vertices of the rectangular disc ($\Delta_y \pm L_y$, $\Delta_z \pm L_z$) by changing the signs $s_y$ and $s_z$ between $-1$ and 1. Because of symmetry, the underlined terms in Eq. (24) vanish when summed over the four vertices of the rectangular disc with the signs $-s_ys_z$. As such, we can define a simplified version of $\Omega$ as

$$\Omega^*(h, a, b)$$
$$= \frac{-1}{2\pi}\left(\arctan(a + b/h) + \arctan\left(\frac{h + ab + a^2h}{b}\right)\right)$$
$$+ T(h, a + b/h) + T\left(\frac{b}{\sqrt{1+a^2}}, \frac{h + ab + a^2h}{b}\right), \tag{28}$$

and hence, by replacing Eq. (24) with Eq. (28), the rotor-averaged deficit of the rectangular disc is

$$\frac{\overline{W}_r^{(n)}}{C} = \left(\frac{\pi\sigma^2\sqrt{1-\xi^2}}{2nL_yL_z}\sum_{s_y, s_z \in \{-1,1\}} (-s_ys_z)\right.$$
$$\left.\Omega^*\left(\frac{\Delta_z + s_zL_z}{\sigma/\sqrt{n}}, \frac{\omega}{\sqrt{1-\xi^2}}, \frac{\Delta_y + s_yL_y}{\sigma\sqrt{(1-\xi^2)/n}}\right)\right)^{1/n}. \tag{29}$$

It should be noted that the two arctan functions in Eq. (28) can be combined into $\arctan(1/a)$. However, determining the proper quadrant would require evaluating the original arguments (i.e. $a + b/h$ and $h/b + a + a^2h/b$), and hence Eq. (28) can be simply used in its current format. The functions T and $\Omega$ appear in the solution of the rectangular disc solely due to having $\omega > 0$ (i.e. due to wind veer). In the case of no wind veer ($\omega = 0$), the rotor-averaged deficit for the rectangular

disc simplifies to

$$\left.\frac{\overline{W}_r^{(n)}}{C}\right|_{\omega=0} = \left(\frac{\pi\sigma^2\sqrt{1-\xi^2}}{8nL_yL_z}\left[\text{erf}\left(\frac{\Delta_y + L_y}{\sigma\sqrt{2(1-\xi^2)/n}}\right)\right.\right.$$
$$\left.- \text{erf}\left(\frac{\Delta_y - L_y}{\sigma\sqrt{2(1-\xi^2)/n}}\right)\right]\left[\text{erf}\left(\frac{\Delta_z + L_z}{\sigma\sqrt{2/n}}\right)\right.$$
$$\left.\left.- \text{erf}\left(\frac{\Delta_z - L_z}{\sigma\sqrt{2/n}}\right)\right]\right)^{1/n}. \tag{30}$$

Furthermore, for the specific case of axisymmetric wake ($\omega = \xi = 0$), the rotor-averaged deficit for the rectangular disc becomes

$$\left.\frac{\overline{W}_r^{(n)}}{C}\right|_{\omega=\xi=0} = \left(\frac{\pi\sigma^2}{8nL_yL_z}\left[\text{erf}\left(\frac{\Delta_y + L_y}{\sigma\sqrt{2/n}}\right)\right.\right.$$
$$\left.- \text{erf}\left(\frac{\Delta_y - L_y}{\sigma\sqrt{2/n}}\right)\right]\left[\text{erf}\left(\frac{\Delta_z + L_z}{\sigma\sqrt{2/n}}\right)\right.$$
$$\left.\left.- \text{erf}\left(\frac{\Delta_z - L_z}{\sigma\sqrt{2/n}}\right)\right]\right)^{1/n}. \tag{31}$$

What remains here is to find the size of the rectangular disc ($L_y$ and $L_z$). It is not straightforward to obtain a mathematically exact expression for the size of the rectangular disc ($L_y$ and $L_z$) that makes Eq. (29) match the case of a circular disc exactly. However, we can compare the simplified linear solutions ($n = 1$) of both cases for an axisymmetric wake ($\omega = \xi = 0$) with no offset ($\rho = 0$) and no hub-height difference ($\delta = 0$), just to have a rough estimate of $L_y$ and $L_z$. We also simplify the rectangular disc to a square and assume that $L_y = L_z = L$. By doing so, Eqs. (7) and (31) simplify to

$$\frac{2\sigma^2}{R^2}\left(1 - \exp\left(\frac{-R^2}{2\sigma^2}\right)\right) = \frac{\pi\sigma^2}{2L^2}\text{erf}^2\left(\frac{L}{\sigma\sqrt{2}}\right). \tag{32}$$

An approximate solution to Eq. (32) takes the form

$$\frac{L}{R} \approx \left(\frac{\sqrt{\pi}}{2}\right)^{\text{erf}(2\sigma/R)}, \tag{33}$$

which can be further simplified by realising that for a typical inter-turbine spacing $2\sigma/R \gg 1$ TS9, leading to $\text{erf}(2\sigma/R) \sim 1$. Therefore, an approximate simpler form for the rectangular-disc size is

$$L_y = L_z \approx \frac{\sqrt{\pi}}{2}R, \tag{34}$$

where $R$ is the turbine's radius. Although this is a simplified analysis conducted under many restrictions (e.g. axisymmetric wake), we will show in Sect. 3 that Eq. (34) gives good agreement with numerical solutions for a wide range of wake parameters. Additionally, Eq. (34) is equivalent to equating the surface area of the circular and rectangular discs, similar to Cheung et al. (2024). A Python implementation of Eq. (29) is available from Ali et al. (2024c).

## 3 Verification, compute costs, and uncertainty

In this section, we verify the derived analytical solutions (Eqs. 18 and 29) by comparing them to numerical evaluations of the rotor-averaged deficit. First, we examine the case of a single upstream wake, considering both circular and rectangular discs (Sect. 3.1). The analysis in Sect. 2 shows that deficit contours are influenced by wind veer and also by the averaging order $n$. We investigate the impact of these parameters on the rotor-averaged deficit in Sect. 3.2 and 3.3, respectively. Of less impact on the rotor-averaged deficit is the yaw misalignment of the wake source, which is presented as additional material in Appendix G. Furthermore, we apply the derived analytical solutions (Eqs. 18 and 29) to scenarios with multiple upstream wakes, using various wake superposition models. For numerical reference, the analytical solutions are evaluated against a set of 2000 averaging points uniformly distributed across the rotor disc in a sunflower pattern as illustrated in Fig. E1 (see Appendix E). The computational cost of the derived analytical solutions compared to numerical averaging at various resolutions is presented in Sect. 3.5. Finally, the uncertainty in predicting the rotor-averaged deficit for the proposed analytical solutions and for various numerical resolutions are quantified compared to 2000-point averaging in Sect. 3.6.

### 3.1 Single upstream wake

We consider the non-axisymmetric Gaussian wake of a wind turbine (Eq. 1) and compute the linear rotor-averaged deficit ($n = 1$) experienced by a downstream turbine modelled as a circular disc and as a rectangular disc at various downstream distances relative to the wake source.

For this analysis, the upstream turbine (wake source) is configured to operate at a yaw misalignment $\gamma_o = 20°$ and a thrust coefficient $C_t = 0.8$ in a free-stream turbulence intensity $T_i = 5\%$. The influence of varying the yaw misalignment of the wake source is small compared to veer effects as outlined in Appendix G. In line with the problem formulation in Sect. 2.1, we assume a differential wind direction of $7°$ across the upstream turbine's top and bottom tips, representing moderate wind veer affecting a large turbine (diameter $\sim 220$ m), resulting in a veer coefficient $\omega \approx 0.122 \, x/D_o$. Stronger wind veer is considered in Sect. 3.2. At each downstream position, the wake eccentricity $\xi$ is calculated based on Eqs. (3) and (4) under the assumption of isotropic wake expansion rate normal to the free-stream direction (i.e. $k_y^* = k_z^* = k^*$) using the empirical expression $k^* = 0.003678 + 0.3837 \, T_i$ (Niayifar and Porté-Agel, 2016). Alternative empirical expressions for $k^*$ are available; however, their use does not affect the verification process, as both analytical and numerical solutions depend on $\sigma$ irrespective of how it is defined. We consider here the linear rotor-averaged deficit ($n = 1$), given the relatively high values of yaw misalignment ($\gamma_o = 20°$) and wind veer ($\Delta\alpha_o = 7°$) in

this set-up. Increasing the averaging order $n$ here would extend the derived expression for a circular disc (Eq. 18) beyond the moderate wake shearing and stretching assumptions under which Eq. (18) was developed. Higher averaging orders can, however, be explored with lower wind veer or with a rectangular disc as discussed in more detail in Sect. 3.3.

Figure 2 illustrates the normalised linear rotor-averaged wind-speed deficit for a circular disc (dashed; Eq. 18) and for a rectangular disc (solid; Eq. 29) as a function of the offset variation ($\rho/\sigma$) at different downstream locations, compared to numerical results (markers; based on points in Fig. E1) across several values of the angle $\delta$. For each case, the values of the eccentricity $\xi$ (Eq. 3), veer coefficient $\omega$ (Eq. 2), ratio $R/\sigma$ (Eq. 4), and skewness parameter $\kappa^{(1)}$ (Eq. 14) are specified. During the derivation of Eq. (18), the skewness parameter $\kappa^{(n)}$ was assumed to remain sufficiently small ($\lesssim 1$) to enable the approximation $I_\nu \left( n\eta^2 R^2/(2\sigma_{ns}^2) \right) \sim (n\eta^2 R^2/(4\sigma_{ns}^2))^\nu/\nu!$ (Appendix C). The $\kappa^{(1)}$ values shown in Fig. 2 verify this assumption, with the maximum $\kappa^{(1)}$ being approximately 0.27 at 10 diameters downstream of the wake source (Fig. 2d). A practical limit on $\kappa^{(n)}$ so that Eq. (18) maintains high accuracy is approximately 0.4–0.5 as will be outlined in Sect. 3.2 and 3.3.

Comparison with numerical evaluations of the rotor-averaged deficit confirms the high accuracy of Eq. (18), even at far-wake downstream distances ($x/D_o = 10$) where wind veer has significantly sheared the wake. The mean absolute error (difference between analytical and numerical solutions) for the circular-disc solution is approximately $7.2 \times 10^{-3}$, with a maximum error of $22.5 \times 10^{-3}$ occurring in the case of zero offset ($\rho = 0$). The deviations between the circular-disc results and the numerical results at zero offset ($\rho = 0$ in Figs. 2c and d) are primarily related to the simplifying assumption in Sect. 2.2 (Appendix C), where $I_0 \left( n\eta^2 R^2/(2\sigma_{ns}^2) \right) \sim 1$ was employed. However, at large distance downstream of the wake source (10 diameters and further), the scaling function $C$ diminishes enough that these differences are negligible for rotor-averaged deficit evaluation. At zero offset between the considered turbine and the wake centre ($\rho = 0$), the congruence between Eq. (18) and numerical solutions (Fig. 2) across cases indicates that the minimal impact of wake shearing and stretching on the wake centre was a valid assumption. As wake stretching and shearing intensify due to wind veer with increasing downstream distance ($\omega \propto x$; Eq. 2), the influence of the angle $\delta$ (hub-height difference) becomes increasingly important compared to positions closer to the wake source.

As for the case of a rectangular-disc representation of the turbine (Eqs. 29 and 34), the comparison in Fig. 2 (solid curves) shows that the rectangular-disc solution provides an excellent accuracy, performing better than the circular-disc solution without the limitations observed for the circular-disc solution at no offset (e.g. $\rho = 0$ in Fig. 2d). Specifically, the mean error for the rectangular-disc solution is approximately

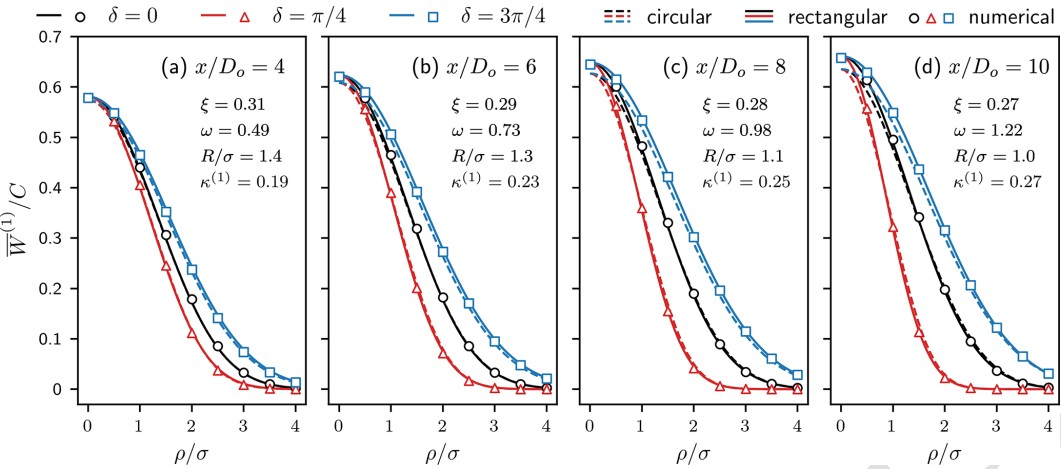

**Figure 2.** Comparing the normalised linear rotor-averaged deficit of a circular disc (dashed; Eq. 18) and a rectangular disc (solid; Eq. 29) against numerical averaging (markers) using the set of discrete points shown in Fig. E1 for different values of the normalised offset $\rho/\sigma$. The upstream turbine operates at a yaw misalignment $\gamma_o = 20°$ and at a thrust coefficient $C_t = 0.8$ in a free-stream turbulence intensity $T_i = 5\%$ with a wind-direction difference $\Delta\alpha_o = 7°$ across its top and bottom tips. Indicated for each downstream location $x/D_o$ are the wind-veer coefficient $\omega \approx 0.122\,x/D_o$ (for $\Delta\alpha_o = 7°$; Eq. 2), the eccentricity of the wake elliptic contour $\xi$ (for $\gamma_o = 20°$; Eqs. 3 and 4), the ratio of the radius of the considered turbine to the wake standard deviation $R/\sigma$ ($\sigma$ is obtained from Eq. 4), and the skewness parameter $\kappa^{(1)}$ (Eq. 14). For each downstream location $x/D_o$, three values of the angle $\delta$, which is the angle between the wake centre and the centre of the considered turbine (Fig. 1), are considered: $0$, $\pi/4$, and $3\pi/4$.

$2.7 \times 10^{-3}$, which is approximately a third of that of the circular-disc solution, with a maximum error of $7.2 \times 10^{-3}$. Besides the higher accuracy, we will show in Sect. 3.2 and 3.3 that the rectangular-disc solution offers further advantages over the circular-disc solution by consistently predicting the rotor-averaged deficit with higher accuracy in cases of significant wind veer and/or higher averaging orders, scenarios in which the circular-disc solution is less accurate due to an elevated skewness parameter $\kappa^{(n)}$.

## 3.2 Effect of wind veer

In the comparisons shown thus far (Fig. 2), a wind-direction difference of $\Delta\alpha_o = 7°$ was set across the top and bottom tips of the wake source, reflecting a moderate veer acting on a large upstream turbine (Sect. 2.1). The circular-disc solution (Eq. 18) was derived based on the assumption of small or moderate veer, which implies a small skewness parameter ($\kappa^{(n)}$; Eq. 14). Here, we examine a range of wind-veer magnitudes by varying the wind-direction difference $\Delta\alpha_o$ for both circular and rectangular discs to evaluate the accuracy of each under conditions of low to high wind veer.

Figure 3 presents the linear rotor-averaged deficit for both circular (dashed curves; Eq. 18) and rectangular (solid curves; Eq. 29) disc models, compared against numerical averaging (markers). In this set-up, the upstream turbine (wake source) has $C_t = 0.8$ and $T_i = 5\%$, as before, but with zero yaw ($\gamma_o = 0°$) to isolate the impact of wind veer. Both the upstream and the considered turbines have the same hub height (i.e. $\delta = 0$). For small wind veer ($\Delta\alpha_o = 5°$, black curves in Fig. 3), both the circular- and rectangular-disc solutions match the numerical solutions with high accuracy with a maximum error of $8.1 \times 10^{-3}$ for the circular disc and $5.2 \times 10^{-3}$ for the rectangular disc.

The advantage of the rectangular-disc solution becomes evident for the case of moderate wind veer ($\Delta\alpha_o = 15°$, red curves in Fig. 3), where the rectangular-disc solution matches the numerical one at all downstream locations, whereas the circular-disc solution deviates from the numerical solution with the streamwise distance (e.g. dashed red curve in Fig. 3d). Specifically, the circular disc has mean and maximum errors of $4.5 \times 10^{-2}$ and $10^{-1}$, respectively, which is 1–2 orders magnitude higher than the errors in the case of $\Delta\alpha_o = 5°$. Conversely, the rectangular-disc solution maintains higher accuracy with mean and maximum errors of $3.9 \times 10^{-3}$ and $10^{-2}$, respectively.

In cases of strong veer ($\Delta\alpha_o = 45°$), only the rectangular-disc solution (Eq. 29) remains valid, as the circular-disc solution (Eq. 18) fails due to the high skewness parameter ($\kappa^{(1)} > 2$), which violates the underlying assumptions of Eq. (18) (see Sect. 2.2 and Appendix C for details). Nevertheless, the rectangular-disc solution continues to yield pre-

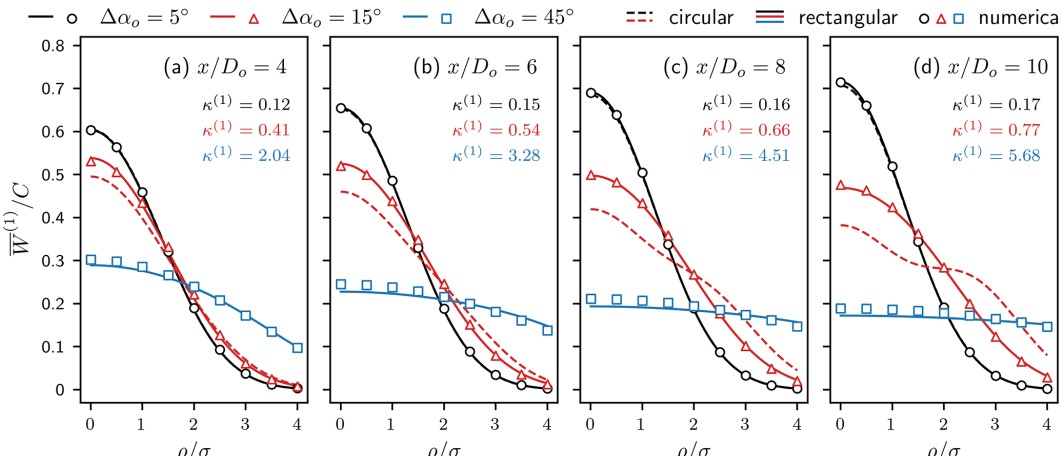

**Figure 3.** Comparing the analytical and numerical linear rotor-averaged deficit for different values of wind veer by changing the wind-direction difference $\Delta\alpha_o$ across the top and bottom tips of the upstream turbine (wake source). The analytical solutions shown are that of a circular disc (dashed; Eq. 18) and a rectangular disc (solid; Eq. 29), whereas numerical averaging (markers) is obtained using the averaging points of Fig. E1. Similar to Fig. 3.1, the upstream turbine has a thrust coefficient $C_t = 0.8$ and operates in a free-stream turbulence intensity $T_i = 5\%$ but has no yaw (i.e. $\gamma_o = 0$). The skewness parameter $\kappa^{(1)}$ for each veer case is indicated, and $\delta = 0$ (same hub height) for all cases.

dictions of rotor-averaged deficit that are consistent with numerical solutions even under extreme veer conditions within a neutral boundary layer. Of slightly less accuracy than the lower veer cases, the mean and maximum errors for the rectangular disc are $9 \times 10^{-3}$ and $1.7 \times 10^{-2}$, respectively, which are more accurate than the circular disc results at $\Delta\alpha_o = 15°$. The larger error for this case ($\Delta\alpha_o = 45°$) compared to the previous two cases is primarily due to the empirical expression for the size of the rectangular disc (Eq. 34). Higher accuracy could be achieved if the dimensions of the rectangular disc are optimised, even though current accuracy is acceptable.

The results in Fig. 3 indicate that for the circular-disc solution, a practical limit for the skewness parameter $\kappa^{(n)}$ would be around 0.4, beyond which the circular-disc solution deviates from the numerical solution. Generally, as wind veer increases (i.e. as shearing of deficit contours intensifies), the deficit contours take on the appearance of a horizontally oriented strip of non-zero deficit. This trend is evident in Fig. 3d for $\Delta\alpha_o = 45°$, where the rotor-averaged deficit remains approximately constant with respect to the offset $\rho$. Although this extreme case was analysed to test the limits of the analytical solutions, it is unlikely to be encountered in a neutral boundary layer where the Gaussian wake model (Eq. 1) applies.

### 3.3 Effect of the averaging order

Besides wind veer, the averaging order $n$ has a direct influence on the skewness parameter $\kappa^{(n)}$ (Eq. 14) and hence on the shearing of the deficit contours. Here, we examine the circular and rectangular solutions for different averaging orders $n$ by comparing them to numerical rotor averaging as presented in Fig. 4. The wake source operates at yaw misalignment $\gamma_o = 20°$ (effect of yawing the wake source is minimal as shown in Appendix G) and at $C_t = 0.8$ in a free-stream turbulence intensity of $5\%$. The wind-direction difference across the tips of the wake source $\Delta\alpha_o = 7°$ (Eq. 2), and both turbines have the same hub height ($\delta = 0$).

The case of linear averaging ($n = 1$; black in Fig. 4) was already examined in previous sections, where both the circular (dashed curves) and rectangular (solid curves) solutions agree well with the numerical solution (markers). However, increasing the averaging order results in accuracy deterioration of the circular-disc solution, especially at larger distances downstream (e.g. Fig 4c, d). For instance, the circular-disc solution deviated significantly from the numerical solution for the cubic averaging ($n = 3$) at almost all downstream distances. Conversely, the rectangular-disc solution (Eq. 29) has excellent agreement with the numerical solution at all distances and all averaging orders, highlighting its robustness and accuracy over the circular-disc solution.

The impact of the averaging order $n$ on the rotor-averaged deficit (analytical or numerical) is not trivial, as indicated by Fig. 4. However, assessing the accuracy of each averaging order is out of the scope of the current study and should be conducted by comparing different averaging orders to a higher-fidelity model.

### 3.4 Multiple upstream wakes

So far, we have examined the analytical solutions derived for a single upstream wake (Eqs. 18 and 29). However, in real applications, a wind turbine is often influenced by multiple upstream turbines, demanding the use of wake superposition models. When numerically calculating the rotor-

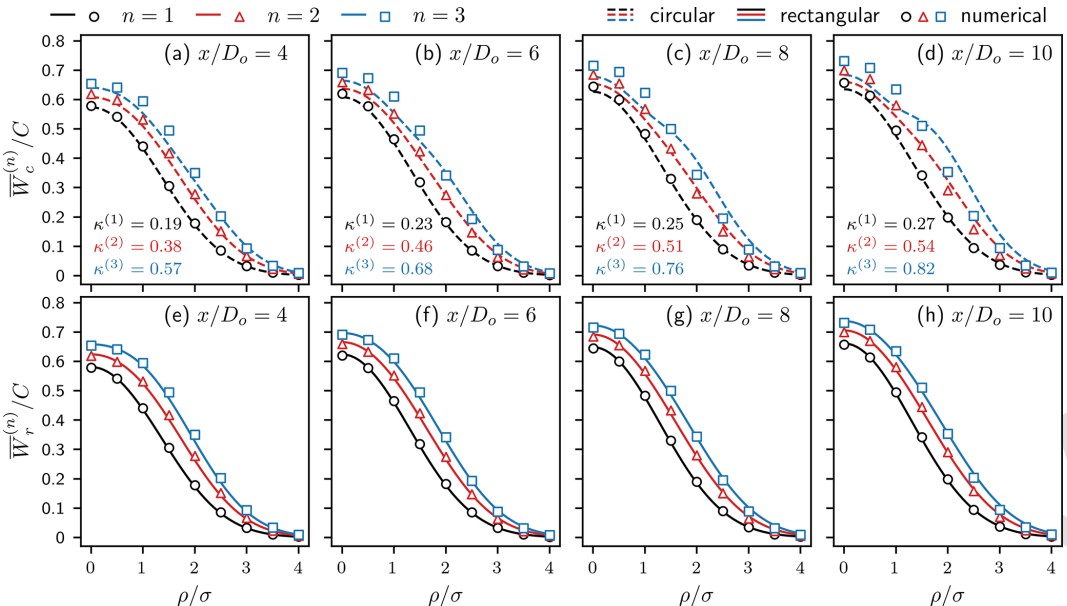

**Figure 4.** Comparison of the analytical (circular and rectangular) rotor-averaged deficit to numerical averaging for different averaging orders $n$. The top row corresponds to the circular-disc solution (Eq. 18), whereas the bottom row is the rectangular-disc solution (Eq. 29). The upstream turbine (wake source) operates at a yaw misalignment $\gamma_o = 20°$ at $C_t = 0.8$ in a free-stream turbulence intensity $T_i = 5\%$. The wind-direction difference across the top and bottom tips of the wake source $\Delta\alpha_o = 7°$, and both turbines have the same hub height (i.e. $\delta = 0$). The skewness parameter $\kappa^{(n)}$ (Eq. 14) is indicated for each case.

averaged deficit from multiple upstream wakes, a discrete set of points over the rotor disc are used. At each point, wake superposition is applied for all upstream wakes individually, and the rotor-averaged deficit is determined from these super-
5  posed deficits. Alternatively, in the analytical approach, the rotor-averaged deficit is calculated for each upstream wake independently (using Eqs. 18 or 29) before superposing the rotor-averaged deficits. The effect of the sequence in which wake superposition and rotor averaging are applied depends
10  on the structure of the superposition model. As shown by Ali et al. (2024a) for axisymmetric wakes, the order of superposition and rotor averaging has minimal influence on the overall rotor-averaged deficit, regardless of whether linear superposition (Niayifar and Porté-Agel, 2016) or root-mean-square
15  (rms) superposition (Voutsinas et al., 1990) is used, and they demonstrated this by application to the Horns Rev wind farm. In this section, we extend the analysis to non-axisymmetric wakes to assess the impact of the order of wake superposition and rotor averaging.
20  Expanding on the work by Ali et al. (2024a), we analyse the Horns Rev wind farm but with yawed turbines to evaluate the accuracy of Eqs. (18) and (29) when combined with different wake superposition models compared to numerical approaches. The yaw misalignment of each turbine is based on
25  the optimisation study by Zhang et al. (2024), where a free-stream wind blows from west to east at a speed of $8\,\mathrm{m\,s^{-1}}$ and a turbulence intensity of $7.7\%$. Figure 5a shows the row-averaged optimised yaw misalignment of the Horns Rev

wind farm along with a schematic of the farm's layout and wind direction. We use the wake deflection model from Bas-  30
tankhah and Porté-Agel (2016) and the turbine-added turbulence model by Crespo and Hernandez (1996), assuming each turbine's wake has a Gaussian shape consistent with Eq. (1). Wind-veer effects are excluded in this comparison (i.e. $\omega = 0$), and all rotor-averaged deficits are linear (i.e.  35
$n = 1$), so the superscript (1) is omitted for brevity.

We consider three wake superposition models: linear superposition (Niayifar and Porté-Agel, 2016), root-mean-square superposition (Voutsinas et al., 1990, hereafter rms), and the product-based superposition by Lanzilao and Meyers  40
(2022). These models are expressed as

$$\overline{W}_{\mathrm{lin}} = \frac{1}{U_\infty}\sum_{j\in S}\overline{u}_j\,\overline{W}_j, \quad \overline{W}_{\mathrm{rms}} = \frac{1}{U_\infty}\sqrt{\sum_{j\in S}\overline{u}_j^2\,\overline{W}_j^2},$$

$$\text{and } \overline{W}_{\mathrm{prod}} = 1 - \prod_{j\in S}\left(1 - \overline{W}_j\right), \tag{35}$$

where $U_\infty$ denotes the free-stream wind speed, $S$ is the set of upstream turbines influencing the considered turbine, and $\overline{u}_j$ and $\overline{W}_j$ represent the rotor-averaged wind speed and  45
the rotor-averaged deficit of a turbine of index $j$. Equations (35) follow the analytical approach in which each upstream wake's rotor-averaged deficit is computed first, followed by superposition. In contrast, the numerical approach applies wake superposition across all upstream wakes before  50
rotor averaging, as will be further discussed in Sect. 4. Following Zhang et al. (2024), the power generation of a turbine

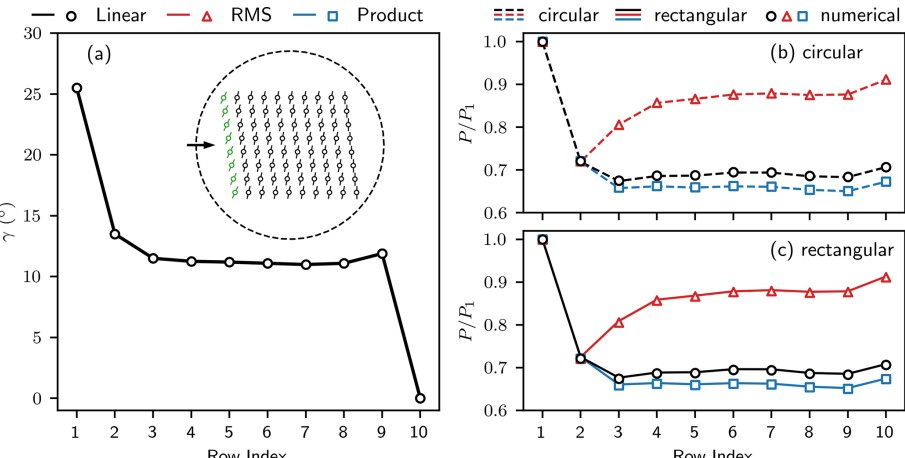

**Figure 5. (a)** The row-averaged yaw misalignment of the Horns Rev wind farm with respect to a free-stream wind from west to east (Zhang et al., 2024). Inset shows a schematic of the farm's layout where the yaw of each turbine is indicated, and the first row is highlighted in green to indicate row definition. **(b)** The row-averaged normalised power generation (in reference to first row) of the Horns Rev wind farm using linear wake superposition (Niayifar and Porté-Agel, 2016, black), root-mean-square superposition (Voutsinas et al., 1990, red), and the product-based wake superposition model (Lanzilao and Meyers, 2022, blue). Power generation (Eq. 36) is obtained using linear averaging of a circular disc (Eq. 18) and is compared to a numerical solution using the averaging points shown in Fig. E1 (markers). The free-stream wind speed is $8\,\mathrm{m\,s^{-1}}$, and the free-stream turbulence intensity is $7.7\,\%$. **(c)** The same as in **(b)** but for the rectangular-disc solution (Eq. 29).

of index $k$ with yaw misalignment $\gamma_k$ is given by

$$P_k = P(\overline{u}_k)\cos^{1.8}(\gamma_k), \tag{36}$$

where $P(u)$ is based on the power-generation table of the Vestas V80-2.0 turbine (used in Horns Rev). Alternative methods to calculate power under yaw include modifying the power coefficient instead of absolute power (similar to Eq. 36), but this section focuses on a unified comparison framework for analytical and numerical solutions, regardless of the power calculation method.

Figure 5b and c illustrate the row-averaged power generation in the Horns Rev farm, calculated analytically (circular and rectangular solutions) and numerically (markers) for each superposition model: linear (black), rms (red), and product-based (blue). The row-averaged power due to yaw follows a similar trend to that observed by Zhang et al. (2024), with reduced power in the first row, increased power in the second and third rows, and only minor variations (1 %– 2 %) in subsequent rows. This pattern is consistent across all three superposition models, though later rows (third and beyond) show greater sensitivity to the chosen superposition model than to yaw. The comparisons in Fig. 5b and c demonstrate that the analytical and numerical row-averaged power calculations are nearly identical for both circular and rectangular solutions. This result indicates that the order of wake superposition and rotor averaging does not affect the accuracy of Eqs. (18) and (29), making these equations suitable for use with the considered superposition models as well as any model with similar operators (linear, rms, or product-based). This is further discussed in Sect. 4.

## 3.5 Computational cost

To evaluate the computational efficiency of the derived analytical solutions (Eqs. 18 and 29) in comparison to the numerical calculation of the rotor-averaged deficit, we consider the power generation of a $25 \times 25$ wind farm. The specific conditions of the free-stream flow and turbine set-up are irrelevant here, as the primary objective is to quantify computational costs. Numerical averaging was conducted using vectorised calculations at various resolutions, ranging from 16 to 2000 points.

Table 1 presents the percentage change in computational cost for the analytical solutions and for numerical averaging at different resolutions relative to using 16 averaging points, a common resolution from the literature. Notably, the rectangular-disc analytical solution (Eq. 29) demonstrates a computational speed-up of approximately $10\,\%$ compared to the 16-point numerical reference, making it the only approach that outperforms the baseline. Conversely, the circular-disc solution (Eq. 18) incurs a computational cost approximately $15\,\%$ higher than the 16-point case, rendering its cost comparable to using 80 averaging points.

## 3.6 Uncertainty quantification

Here, we consider various resolutions and distributions of averaging points to quantify the uncertainty that arises when evaluating the rotor-averaged deficit compared to the 2000-point resolution shown in Fig. E1. The considered cases are the derived analytical solutions (Eqs. 18 and 29) and the distribution of averaging points shown in Fig. E2, which include the 16-point quadrature (Eq. E2) of Holoborodko

**Table 1.** Comparing the relative change in computational cost for the derived analytical solutions and for various numerical resolutions in reference to the cost of a numerical evaluation of the rotor-averaged deficit using 16 averaging points. If the computational cost of a specific experiment is $t$, the relative change is calculated as $(t - t_{16})/t_{16} \times 100\%$, where $t_{16}$ is the computational cost of numerically averaging 16 points using vectorised calculations.

| No. points | Rect. | 50 | Circle | 100 | 500 | 2000 |
|---|---|---|---|---|---|---|
| Relative change | $-10.4\%$ | $6.7\%$ | $14.9\%$ | $21.1\%$ | $112.4\%$ | $443.8\%$ |

(2011, hereafter, Q16), the 16-point cross-like distribution of Stipa et al. (2024, hereafter, C16), and various resolutions of the sunflower distribution (Eq. E1) ranging from 16 (S16) to 1000 (S1000) averaging points. To quantify uncertainty, we calculate the root-mean-square error (RMSE) of each approach (analytical or numerical) against the 2000-point reference case (Fig. E1), where RMSE is defined as

$$\text{RMSE} = \sqrt{\frac{1}{N_s} \sum_{k=1}^{N_s} \left(\overline{W}_k - \overline{W}_{\text{ref}}\right)^2}, \tag{37}$$

where $N_s$ is the number of tested scenarios (i.e. different combinations of driving parameters such as veer and yaw), $\overline{W}_k$ is the rotor-averaged deficit for a scenario of index $k$, and $\overline{W}_{\text{ref}}$ is the reference rotor-averaged deficit (from 2000 averaging points). Different scenarios are generated through different combinations of the yaw misalignment of the wake source ($\gamma_o$), the wind-direction differential across the wake source ($\Delta\alpha_o$), the averaging order $n$, the normalised streamwise distance $x/D_o$, the angle $\delta$, and the normalised offset $\rho/\sigma$. Based on the analysis in Sect. 3.2 and 3.3 and Appendix G (on the effect of $\gamma_o$), wind veer was shown to have the largest impact on the rotor-averaged deficit. As such, we create two sets of scenarios different from each other only in the range of $\Delta\alpha_o$ as indicated in Table 2 and where the circular-disc solution (Eq. 18) is tested only in the "small–moderate" veer scenario (first row in Table 2) within the range of applicability as established in Sect. 3.2 and 3.3.

Table 3 lists scaled values of RMSE (Eq. 37; scaled by 1000) for the aforementioned cases (analytical and numerical) and wake scenarios. As the number of averaging points increases, it is expected that RMSE drops as the predicted solution converges to that of the reference 2000-point case (e.g. S1000 vs. S100 in Table 3) but at the expense of computational cost as outlined in Sect. 3.5. For the small–moderate veer scenario, the Q16 case has the lowest RMSE ($5.5 \times 10^{-3}$) among the analytical cases and the 16-point averaging cases, with an accuracy that is approximately similar to that of 500 averaging points (S500). For the same scenario, the circular-disc solution has an approximately similar RMSE as 100 averaging points (S100), whereas the rectangular-disc solution has slightly higher accuracy.

The moderate–high veer scenario indicates that the rectangular-disc solution (Eq. 29) has higher accuracy than all 16-point averaging cases at an RMSE of $11.5 \times 10^{-3}$

(similar to the small–moderate veer scenario). In contrast, the accuracy of Q16 is significantly reduced with RMSE of $17.2 \times 10^{-3}$. The averaging distributions C16 and S16 have significantly less accuracy than the other cases, holding the highest RMSE for both veer scenarios. For the moderate–high veer scenario, 100 averaging points provide comparable accuracy to the rectangular-disc solution, whilst 500 averaging points, which are computationally expensive (Table 1), are required to have the same accuracy of Q16 in the small–moderate veer scenario.

## 4 Discussion

The presented solutions in Sect. 2 are compatible with any wake deflection model from the literature as all distances were referenced to the wake centre. However, if the centre of the upstream turbine is sought to be the reference location, then the definitions of the offset $\rho$ and the angle $\delta$ need modifications to account for the wake horizontal deflection $d_o$. In this case, the modified offset $\rho^*$ and the modified angle $\delta^*$ measured from the centre of the upstream turbine are

$$\rho^* = \rho\sqrt{1 + 2\left(\frac{d_o}{\rho}\right)\cos\delta + \left(\frac{d_o}{\rho}\right)^2},$$

$$\text{and } \tan\delta^* = \frac{\sin\delta}{d_o/\rho + \cos\delta}. \tag{38}$$

The expressions for the rotor-averaged deficit (Eqs. 18 and 29) were derived for a generic averaging order $n > 0$, where the case of $n = 1$ is equivalent to obtaining the averaged momentum deficit through the turbine rotor (for incompressible steady flow), $n = 2$ corresponds to the averaged kinetic-energy deficit through the rotor, and $n = 3$ is equivalent to the averaged power deficit through the rotor. To obtain a solution for a circular disc (Eq. 18), it was assumed that the stretching and shearing of the wake contours are not large as those quantified by the skewness parameter $\kappa^{(n)}$ (Eq. 14). As such, using higher-order averaging for a circular disc should be limited to cases of small or moderate wind veer (if any) to keep Eq. (18) within its validity region. Conversely, the solution of the rectangular disc (Eq. 29) is not limited by this simplifying assumption and was shown to perform well even in the case of extreme wind veer (Fig. 3), giving it a large advantage against the circular-disc solution.

**Table 2.** Ranges of the driving parameters considered in uncertainty quantification scenarios. The angle $\gamma_o$ is the yaw misalignment of the wake source, $\Delta\alpha_o$ is the wind-direction differential across the top and bottom tips of the wake source, $n$ is the averaging order, $x/D_o$ is the normalised streamwise distance measured from the wake source, $\delta$ is the azimuthal coordinate of the considered turbine centre measured from the wake centre (Fig.1), and $\rho/\sigma$ is the normalised offset between the wake centre and the centre of the considered turbine (Fig. 1), where $\sigma$ is the wake standard deviation (Eq. 4). Ranges written in the form $v_o : v_s : v_f$ mean this variable ranges from $v_o$ to $v_f$ (inclusive) with a step of $v_s$.

| Veer scenario | $\gamma_o$ | $\Delta\alpha_o$ | $n$ | $x/D_o$ | $\delta$ | $\rho/\sigma$ |
|---|---|---|---|---|---|---|
| Small–moderate | $0° : 10° : 30°$ | $0° : 1° : 7°$ | $\{1, 2, 3\}$ | $4 : 2 : 10$ | $0 : \pi/4 : 3\pi/4$ | $0 : 0.5 : 4$ |
| Moderate–high | $0° : 10° : 30°$ | $10° : 5° : 45°$ | $\{1, 2, 3\}$ | $4 : 2 : 10$ | $0 : \pi/4 : 3\pi/4$ | $0 : 0.5 : 4$ |

**Table 3.** Scaled root-mean-square error ($1000 \times$ RMSE; Eq. 37) of different rotor averaging cases including the rectangular-disc solution (Rect.; Eq. 29), the circular-disc solution (Circle; Eq. 18), the 16-point quadrature (Q16; Eq. E2) shown in Fig. E2a, the 16-point cross-like distribution shown in Fig. E2b, and various resolutions of the sunflower distribution (starting with S; Eq. E1) ranging from 16 averaging points (S16) to 1000 averaging points (S1000). The reference to which each case is compared is numerical averaging using 2000 points following a sunflower distribution as indicated in Fig. E1. The ranges of the driving parameters for both veer scenarios are listed in Table 2. The abbreviation n/a stands for not applicable.

| Veer scenario | Rect. | Circle | Q16 | C16 | S16 | S100 | S500 | S1000 |
|---|---|---|---|---|---|---|---|---|
| Small–moderate | 10.0 | 11.4 | 5.5 | 33.8 | 22.0 | 11.2 | 6.0 | 3.8 |
| Moderate–high | 11.5 | n/a | 17.2 | 42.2 | 27.4 | 12.6 | 6.4 | 4.1 |

The analytical solutions proposed in Sect. 2.2 and 2.3 (Eqs. 18 and 29) correspond to a single upstream wake, whereas an operational wind turbine is typically impacted by multiple upstream wakes whose deficits are superposed using a variety of wake superposition models. For a super-position model that relies on a linear operator to combine upstream deficits (e.g. Lissaman, 1979; Niayifar and Porté-Agel, 2016; Zong and Porté-Agel, 2021; Dar and Porté-Agel, 2024), the numerical and analytical approaches are the same, meaning that the order of applying wake superposition and rotor averaging has no effect. However, other wake su-perposition models rely on root-mean-square operators (e.g. Katic et al., 1987; Voutsinas et al., 1990) for which the or-der of wake superposition and rotor averaging is not triv-ial. Ali et al. (2024a) showed mathematically that for a col-umn of turbines of the same hub height ($\delta = 0$) with no off-set ($\rho = 0$) where the wake of each turbine is axisymmetric ($\xi = \omega = 0$), the order in which wake superposition and rotor averaging are applied results in insignificant differences as long as the number of upstream turbines with non-negligible deficits acting on the considered turbine is not large. They showed that for an analytical approach (rotor-averaging fol-lowed by superposition), the rotor-averaged deficit of the considered turbine is proportional to $\exp(-\tilde{\sigma}^{-2}/4)$, where $\tilde{\sigma}$ is a deficit-weighted averaged wake standard deviation for all upstream turbines impacting the considered turbine, whereas for a numerical approach (superposition followed by rotor-averaging), the rotor-averaged deficit is proportional to $\exp(-2\tilde{\sigma}^{-2}/9)$. In a typical wind farm, the number of up-stream turbines with non-negligible deficits acting on a tur-bine is 2–3, where one of these turbines has the dominant

wake effect, making these two exponents very close. Their conclusion can be easily extended to any averaging order $n$ using the substitution $\tilde{\sigma}^2 \to \tilde{\sigma}^2/n$, and it also naturally ex-tends to the considered case of a non-axisymmetric wake, as the non-axisymmetric solution was shown to be a perturba-tion to a scaled axisymmetric solution (Eq. 18). Application to the Horns Rev wind farm showed that the numerical and analytical approaches using root-mean-square superposition gave indistinguishable results (Figs. 5b and 5c), where all upstream wakes for a specific turbine were considered in the evaluation of the turbine's operating point.

The superposition model of Lanzilao and Meyers (2022) uses neither linear nor root-mean-square operators but rather the product of the normalised rotor-averaged wind speeds of all upstream wake sources. We show in Appendix F that for this superposition model and for any other superposi-tion model of a similar operator, the numerical and analyt-ical approaches are asymptotically identical if the upstream wakes of the considered turbine are assumed to operate in-dependently. This assumption is justified as each turbine can be yawed independently of the other turbines depending on its onset wind, though such a strategy is not optimal for the whole wind-farm performance. We also demonstrate that for small-enough upstream deficits ($W \lesssim 0.3$), this product-based superposition model converges to a non-weighted lin-ear superposition model, which explains the closeness in the estimated power generation by the two wake superposition models when applied to the Horns Rev wind farm (Fig. 5b and c). Similar to root-mean-square superposition, when this product-based superposition model was applied to the Horns

Rev wind farm, there were no distinguishable differences between the analytical and numerical solutions (Fig. 5).

Some limitations should, however, be considered. The rotor-averaging process inherently assumes that a zero-deficit point on the rotor disc has a wind speed that is equal to that of the upstream turbine (wake source) rather than the free-stream wind speed or another background wind speed. This can have profound impacts on the rotor-averaged wind speed in the case of highly heterogeneous flow within a wind farm, such as in the case of hurricanes or extremely large wind farms. In such a scenario, all numerical and analytical approaches based on engineering wake models have shortcomings, as the underlying assumptions of the wake-deficit model cannot predict the interactions between the wakes and the heterogeneous background flow.

In some wind-energy applications, the nacelle wind-speed deficit (hub-height deficit) is used as a proxy for the wind speed across the entire rotor. In Appendix G, we compared rotor averaging of the deficit with the nacelle-point deficit (see Fig. G1), indicating that the nacelle deficit can be significantly different from a rotor-averaged value, which could impair the accuracy of estimating a turbine's operating point. Hence, it is recommended to use a rotor-averaged value for the deficit rather than the nacelle-point deficit. We also explored the impact of yawing the wake source on the rotor-averaged deficit of the considered turbine, which was shown to be much less than other parameters such as wind veer and the averaging order. Although not addressed in this study, Eqs. (18) and (29) are differentiable, which allows for obtaining mathematical expressions for the gradients of the rotor-averaged wind speed of a turbine with respect to its location in a farm and/or to the operating point of upstream turbines.

## 5 Summary

In the current study, we derived and verified two expressions for the surface integration of a non-axisymmetric Gaussian wake over a circular disc and an equivalent rectangular disc, depicting the rotor of a turbine whose rotor-averaged deficit is sought. The general integrated wake profile took into consideration wake stretching arising from the yawing of upstream turbines and wake planar shearing due to wind-veer effects through a set of controlling variables as detailed in Sect. 2.1. The presented expressions were verified against numerical evaluations of the rotor-averaged deficit, indicating good agreement for the circular-disc case and excellent agreement for the rectangular case.

While the circular-disc solution matched the numerical evaluations of the rotor-averaged deficit well in the cases of low or moderate wind veer and small averaging order ($n = 1$), the solution's validity range is limited due to some simplifying assumptions made during the derivation of the solution (Sect. 2.2). Conversely, the rectangular disc was not limited to these simplifying assumptions and outperformed the circular-disc solution, especially for the cases of high wind veer and/or large averaging order. In terms of computational cost, both analytical solutions were comparable to vectorised calculations of the rotor-averaged deficit using 16 averaging points, where the rectangular-disc solution was approximately 10 % faster and the circular-disc solution was approximately 15 % slower.

We examined the accuracy of the derived analytical solutions and of various numerical resolutions and distributions of averaging points against high-resolution averaging (using 2000 points). For the same resolution (16 points), we found that the quadrature distribution (Fig. E2a) has significantly higher accuracy than the cross-like distribution (Fig. E2b) and higher than a random distribution of the same number of points (depicted by the sunflower distribution in Fig. E2c), regardless of the intensity of deficit-contour shearing and stretching. The rectangular-disc solution showed high accuracy for both low- and high-veer scenarios, with a performance equivalent to numerical averaging using approximately 100 points. Additionally, the rectangular-disc solution outperformed quadrature averaging using 16 averaging points in the high-veer scenario.

The expressions of the rotor-averaged deficit for a single turbine wake can be applied to multiple wakes using any available superposition model that relies on linear operators, root-mean-square operators, or product operators, as demonstrated by application to the Horns Rev wind farm with optimised yaw misalignment for each turbine. Whilst not derived in this study, the expressions for the rotor-averaged deficit are differentiable and can be beneficial for optimisation-based applications.

## Appendix A: Transfer of axes for the Gaussian wake equation

In this Appendix, we show how Eq. (7) can be transferred from the wake axes $y'-z'$ to the axes of the considered turbine $y-z$. From Eq. (7) along with the relation between the $y'-z'$ and $y-z$ axes (Fig. 1), $y' = y + \Delta_y$ and $z' = z + \Delta_z$, we have

$$\frac{\overline{W}^{(n)}}{C} = \left( \frac{1}{\pi R^2} \int\limits_0^R \int\limits_0^{2\pi} r \exp\left( \frac{-n(y + \Delta_y + \omega(z + \Delta_z))^2}{2\sigma_y^2} \right) \right.$$
$$\left. \exp\left( \frac{-n(z + \Delta_z)^2}{2\sigma_z^2} \right) d\theta \, dr \right)^{1/n}. \tag{A1}$$

By expanding the brackets in Eq. (A1), the exponents can be written as the product of $\exp\left(-nr^2 c_{r^2}/2\right)$, $\exp\left(-nr\rho c_{r\rho}\right)$, and $\exp\left(-n\rho^2 c_{\rho^2}/2\right)$, where $c_{r^2}$, $c_{r\rho}$, and $c_{\rho^2}$ are coefficients of $r^2$, $r\rho$, and $\rho^2$, respectively. Using $\langle y, z \rangle = r\langle \cos\theta, \sin\theta \rangle$ and $\langle \Delta_y, \Delta_z \rangle = \rho\langle \cos\delta, \sin\delta \rangle$, where $\langle t_1, t_2 \rangle$ means $t_1$ or $t_2$, we have

$$c_{r^2} = \frac{\cos^2\theta}{\sigma_y^2} + \frac{\sin^2\theta}{\sigma_z^2} + \frac{\omega \sin 2\theta}{\sigma_y^2} + \frac{\omega^2 \sin^2\theta}{\sigma_y^2}, \tag{A2}$$

$$c_{r\rho} = \underbrace{\left( \frac{\omega \sin\delta + \cos\delta}{\sigma_y^2} \right)}_{a_1} \cos\theta$$

$$+ \underbrace{\left( \frac{1}{\sigma_z^2} + \omega \left( \frac{\omega \sin\delta + \cos\delta}{\sigma_y^2} \right) \right)}_{a_2} \sin\theta, \qquad (A3)$$

and

$$c_{\rho^2} = \frac{\cos^2\delta}{\sigma_y^2} + \frac{\sin^2\delta}{\sigma_z^2} + \frac{\omega \sin 2\delta}{\sigma_y^2} + \frac{\omega^2 \sin^2\delta}{\sigma_y^2}. \qquad (A4)$$

To simplify Eq. (A2), we use the substitutions $\cos^2\theta = (1 + \cos 2\theta)/2$ and $\sin^2\theta = (1 - \cos 2\theta)/2$:

$$c_{r^2} = \underbrace{\left( \frac{1}{2\sigma_y^2} + \frac{1}{2\sigma_z^2} + \frac{\omega^2}{2\sigma_y^2} \right)}_{1/\sigma_*^2} + \underbrace{\left( \frac{1}{2\sigma_y^2} - \frac{1}{2\sigma_z^2} - \frac{\omega^2}{2\sigma_y^2} \right)}_{1/\sigma_{**}^2}$$

$$\cos 2\theta + \left( \frac{\omega}{\sigma_y^2} \right) \sin 2\theta, \qquad (A5)$$

which can be further simplified by defining $1/\sigma_{ns}^2 = \sqrt{1/\sigma_{**}^4 + \omega^2/\sigma_y^4}$ and $\tan\phi_{ns} = \omega\sigma_{**}^2/\sigma_y^2$ to be

$$c_{r^2} = \frac{1}{\sigma_*^2} + \frac{\cos(2\theta - \phi_{ns})}{\sigma_{ns}^2}. \qquad (A6)$$

Using the same procedure and by replacing $\theta$ with $\delta$, we have

$$c_{\rho^2} = \frac{1}{\sigma_*^2} + \frac{\cos(2\delta - \phi_{ns})}{\sigma_{ns}^2}. \qquad (A7)$$

Finally, Eq. (A3) can be simplified to

$$c_{r\rho} = \cos(\theta - \phi_s)/\sigma_s^2, \qquad (A8)$$

by defining $\sigma_s^2 = 1/\sqrt{a_1^2 + a_2^2}$ and $\tan\phi_s = a_2/a_1$, where $a_1$ and $a_2$ are defined in Eq. (A3).

## Appendix B: Azimuthal integration of non-symmetric Gaussian wake

In this Appendix, we present the solution to the integral $M_\theta$ in Eq. (9), which is an extension to a solution proposed by Gaidash (2023).

$$M_\theta = \frac{1}{\pi} \int_0^{2\pi} \exp\left( \frac{-n\eta^2 R^2 \cos(2\theta - \phi_{ns})}{2\sigma_{ns}^2} \right)$$

$$\exp\left( \frac{-n\eta R\rho \cos(\theta - \phi_s)}{\sigma_s^2} \right) d\theta \qquad (B1)$$

Using the Jacobi–Anger expansion (Abramowitz and Stegun, 1972, 9.1.41–45, p. 361), we can write

$$\exp\left( \frac{-n\eta R\rho \cos(\theta - \phi_s)}{\sigma_s^2} \right) = \sum_{\nu \in \mathbb{Z}} (-1)^\nu I_\nu \left( \frac{n\eta R\rho}{\sigma_s^2} \right)$$

$$\exp\left( i\nu \left[ \theta - \frac{\phi_{ns}}{2} \right] \right) \exp\left( i\nu \left[ \frac{\phi_{ns}}{2} - \phi_s \right] \right), \qquad (B2)$$

where $I_\nu$ is the modified Bessel function of order $\nu$, and $\mathbb{Z}$ is the set of integers. Using Eq. (B2), the integral $M_\theta$ becomes

$$M_\theta = \frac{1}{\pi} \sum_{\nu \in \mathbb{Z}} (-1)^\nu \exp\left( i\nu \left[ \frac{\phi_{ns}}{2} - \phi_s \right] \right) I_\nu \left( \frac{n\eta R\rho}{\sigma_s^2} \right) \int_0^{2\pi}$$

$$\exp\left( \frac{-n\eta^2 R^2 \cos(2\theta - \phi_{ns})}{2\sigma_{ns}^2} \right) \exp\left( i\nu \left[ \theta - \frac{\phi_{ns}}{2} \right] \right) d\theta. \quad (B3)$$

The integral in Eq. (B3) vanishes for odd values of $\nu$. Also, since $M_\theta$ is real we can write

$$M_\theta = 2 \sum_{\nu \in \mathbb{Z}} \cos(\nu(2\phi_s - \phi_{ns})) I_{2\nu} \left( \frac{n\eta R\rho}{\sigma_s^2} \right) \int_0^{\pi}$$

$$\exp\left( \frac{-n\eta^2 R^2 \cos(\theta' - \phi_{ns})}{2\sigma_{ns}^2} \right) \cos(\nu(\theta' - \phi_{ns})) d\theta', \qquad (B4)$$

where $\theta' = 2\theta$. The integral in Eq. (B4) can be solved using (Gradshteyn and Ryzhik, 2007, 3.915(2), p. 491)

$$\int_0^{\pi} \exp(-t \cos\zeta) \cos(\nu\zeta) d\zeta = (-1)^\nu \pi I_\nu(t), \qquad (B5)$$

which is insensitive to a phase shift $\zeta \to \zeta - \phi_{ns}$. Hence, $M_\theta$ becomes

$$M_\theta = 2 \sum_{\nu \in \mathbb{Z}} (-1)^\nu \cos(\nu(2\phi_s - \phi_{ns})) I_{2\nu} \left( \frac{n\eta R\rho}{\sigma_s^2} \right)$$

$$I_\nu \left( \frac{n\eta^2 R^2}{2\sigma_{ns}^2} \right), \qquad (B6)$$

which can be further simplified using the fact that $I_{-\nu}(x) = I_\nu(x)$ for an integer $\nu$:

$$M_\theta = 2 I_0 \left( \frac{n\eta R\rho}{\sigma_s^2} \right) I_0 \left( \frac{n\eta^2 R^2}{2\sigma_{ns}^2} \right) + 4 \sum_{\nu \geq 1} (-1)^\nu$$

$$\cos(\nu(2\phi_s - \phi_{ns})) I_{2\nu} \left( \frac{n\eta R\rho}{\sigma_s^2} \right) I_\nu \left( \frac{n\eta^2 R^2}{2\sigma_{ns}^2} \right). \qquad (B7)$$

## Appendix C: Radial integration of non-axisymmetric Gaussian wake

In this Appendix, we provide a solution to the integral $M_\eta$ defined in Eq. (8) along with the solution of the integral $M_\theta$

(Eq. 11), which was detailed in Appendix B.

$$M_\eta = 2 \int_0^1 \eta \exp\left(\frac{-n\eta^2 R^2}{2\sigma_*^2}\right) I_0\left(\frac{n\eta R\rho}{\sigma_s^2}\right) I_0\left(\frac{n\eta^2 R^2}{2\sigma_{ns}^2}\right) d\eta +$$

$$4\sum_{\nu \geq 1}(-1)^\nu \cos(\nu\phi) \int_0^1 \eta \exp\left(\frac{-n\eta^2 R^2}{2\sigma_*^2}\right)$$

$$I_{2\nu}\left(\frac{n\eta R\rho}{\sigma_s^2}\right) I_\nu\left(\frac{n\eta^2 R^2}{2\sigma_{ns}^2}\right) d\eta \qquad (C1)$$

The argument of the terms in the form $I_\nu\left(n\eta^2 R^2/(2\sigma_{ns}^2)\right)$ is sufficiently small ($\lesssim 1$) for the ranges outlined in Sect. 2.1 and for small values of $n$ with an average value of $nR^2/(6\sigma_{ns}^2)$ for $0 \leq \eta \leq 1$. Hence, we employ the approximation $I_\nu\left(n\eta^2 R^2/(2\sigma_{ns}^2)\right) \sim (n\eta^2 R^2/(4\sigma_{ns}^2))^\nu/\nu!$ (Abramowitz and Stegun, 1972, 9.6.7, p. 375), and the integral $M_\eta$ becomes

$$M_\eta \approx 2\mu_0^{(n)} + 4\sum_{\nu \geq 1}\frac{1}{\nu!}\left(\frac{-nR^2}{4\sigma_{ns}^2}\right)^\nu \cos(\nu\phi)$$

$$\underbrace{\int_0^1 \eta^{1+2\nu} \exp\left(\frac{-n\eta^2 R^2}{2\sigma_*^2}\right) I_{2\nu}\left(\frac{n\eta R\rho}{\sigma_s^2}\right) d\eta}_{\mu_{2\nu}^{(n)}}, \qquad (C2)$$

where

$$\mu_0^{(n)} = \int_0^1 \eta \exp\left(\frac{-n\eta^2 R^2}{2\sigma_*^2}\right) I_0\left(\frac{n\eta R\rho}{\sigma_s^2}\right) d\eta. \qquad (C3)$$

Solving the integral $\mu_0^{(n)}$ is discussed in more detail in Sect. 2.2. The solution to the integrals $\mu_{2\nu}^{(n)}$ is derived in detail in Appendix D.

$$\mu_{2\nu}^{(n)} = \left(\frac{\rho\sigma_*^2}{R\sigma_s^2}\right)^{2\nu}\left[\mu_0^{(n)} - \frac{\sigma_*^2}{nR^2}\exp\left(\frac{-nR^2}{2\sigma_*^2}\right)\right.$$

$$\left.\sum_{k=1}^{2\nu}\left(\frac{\rho\sigma_*^2}{R\sigma_s^2}\right)^{-k} I_k\left(\frac{nR\rho}{\sigma_s^2}\right)\right] \qquad (C4)$$

Therefore, $M_\eta$ becomes

$$M_\eta \approx 2\mu_0^{(n)}\left(1 + 2\sum_{\nu \geq 1}\frac{(-n\chi_{ns}^2)^\nu \cos(\nu\phi)}{\nu!}\right)$$

$$- \frac{4\sigma_*^2}{nR^2}\exp\left(\frac{-nR^2}{2\sigma_*^2}\right)\sum_{\nu \geq 1}\frac{(-n\chi_{ns}^2)^\nu \cos(\nu\phi)}{\nu!}$$

$$\left[\sum_{k=1}^{2\nu}\left(\frac{\rho\sigma_*^2}{R\sigma_s^2}\right)^{-k} I_k\left(\frac{nR\rho}{\sigma_s^2}\right)\right], \qquad (C5)$$

where $\chi_{ns} = \rho\sigma_*^2/(2\sigma_{ns}\sigma_s^2)$. For the sum over $\nu$, we have

$$\mathcal{P}_{ns}^{(n)} = \sum_{\nu \geq 1}\frac{(-n\chi_{ns}^2)^\nu \cos(\nu\phi)}{\nu!}$$

$$= \exp(-n\chi_{ns}^2 \cos\phi)\cos(n\chi_{ns}^2 \sin\phi) - 1. \qquad (C6)$$

Also, the modified Bessel function $I_\nu(x)$ decays rapidly with $\nu$, and hence we only keep the terms with $\nu \leq 2$ in the right-hand side of Eq. (C5). Therefore, Eq. (C5) simplifies to

$$M_\eta \approx 2\mu_0^{(n)}\left(1 + 2\mathcal{P}_{ns}^{(n)}\right) - \frac{4\sigma_*^2 \mathcal{P}_{ns}^{(n)}}{nR^2}\exp\left(\frac{-nR^2}{2\sigma_*^2}\right)$$

$$\left[\frac{\lambda}{\rho}I_1\left(\frac{nR\rho}{\sigma_s^2}\right) + \frac{\lambda^2}{\rho^2}I_2\left(\frac{nR\rho}{\sigma_s^2}\right)\right], \qquad (C7)$$

where $\lambda = R\sigma_s^2/\sigma_*^2$.

## Appendix D: A solution of an integral of the modified Bessel function

In this Appendix, we present a solution to a generic integral in the form

$$\mu_\nu^{(n)}(\beta, \vartheta) = \int_0^1 \eta^{1+\nu}\exp\left(\frac{-n\eta^2\beta^2}{2}\right)I_\nu(n\eta\vartheta)\,d\eta, \qquad (D1)$$

where $\nu$ and $n$ are integers, and $\beta$ and $\vartheta$ are constants. To evaluate $\mu_\nu^{(n)}$, we employ

$$\frac{\partial}{\partial\eta}\left(\eta^\nu I_\nu(n\eta\vartheta)\right) = n\vartheta\eta^\nu I_{\nu-1}(n\eta\vartheta). \qquad (D2)$$

Integrating Eq. (D1) by parts leads to the recursion

$$\mu_\nu^{(n)}(\beta, \vartheta) = -\frac{1}{n\beta^2}\exp\left(\frac{-n\beta^2}{2}\right)I_\nu(n\vartheta) + \frac{\vartheta}{\beta^2}\mu_{\nu-1}^{(n)}(\beta, \vartheta), \qquad (D3)$$

which can be solved using the generating function $\mathcal{F}_n(\eta) = \sum_{\nu \geq 1}\mu_\nu^{(n)}\eta^\nu$. Multiplying Eq. (D3) by $\eta^\nu$ and summing over $\nu$ gives

$$\sum_{\nu \geq 1}\mu_\nu^{(n)}\eta^\nu = -\frac{1}{n\beta^2}\exp\left(\frac{-n\beta^2}{2}\right)\sum_{\nu \geq 1}I_\nu(n\vartheta)\eta^\nu$$

$$+ \frac{\vartheta}{\beta^2}\sum_{\nu \geq 1}\mu_{\nu-1}^{(n)}\eta^\nu, \qquad (D4)$$

which simplifies into

$$\mathcal{F}_n(\eta) = -\frac{1}{n\beta^2}\exp\left(\frac{-n\beta^2}{2}\right)\sum_{\nu \geq 1}I_\nu(n\vartheta)\eta^\nu$$

$$+ \frac{\vartheta\eta}{\beta^2}\left(\mu_0^{(n)} + \mathcal{F}_n(\eta)\right). \qquad (D5)$$

Solving Eq. (D5) for $\mathcal{F}_n(\eta)$ and using Taylor's expansion $(1 - \vartheta\eta/\beta^2)^{-1} = \sum\limits_{m\geq 0}\left(\vartheta\eta/\beta^2\right)^m$,

$$\mathcal{F}_n(\eta) = -\frac{1}{n\beta^2}\exp\left(\frac{-n\beta^2}{2}\right)\sum_{\nu\geq 1}\sum_{m\geq 0}\left(\frac{\vartheta}{\beta^2}\right)^m$$

$$I_\nu(n\vartheta)\eta^{\nu+m} + \mu_0^{(n)}\sum_{m\geq 0}\left(\frac{\vartheta\eta}{\beta^2}\right)^{m+1}. \qquad (D6)$$

TS10 Finally, the integral $\mu_\nu^{(n)}$ is the coefficient of $\eta^\nu$ in Eq. (D6):

$$\mu_\nu^{(n)}(\beta, \vartheta) = \left(\frac{\vartheta}{\beta^2}\right)^\nu\left[\mu_0^{(n)}(\beta, \vartheta)\right.$$

$$\left. - \frac{1}{n\beta^2}\exp\left(\frac{-n\beta^2}{2}\right)\sum_{k=1}^\nu\left(\frac{\vartheta}{\beta^2}\right)^{-k}I_k(n\vartheta)\right], \quad (D7)$$

where $\mu_0^{(n)}(\beta, \vartheta)$ is

$$\mu_0^{(n)}(\beta, \vartheta) = \int\limits_0^1 \eta\exp\left(\frac{-n\eta^2\beta^2}{2}\right)I_0(n\eta\vartheta)\,\mathrm{d}\eta. \qquad (D8)$$

## Appendix E: Distribution of points across a circle for numerical averaging

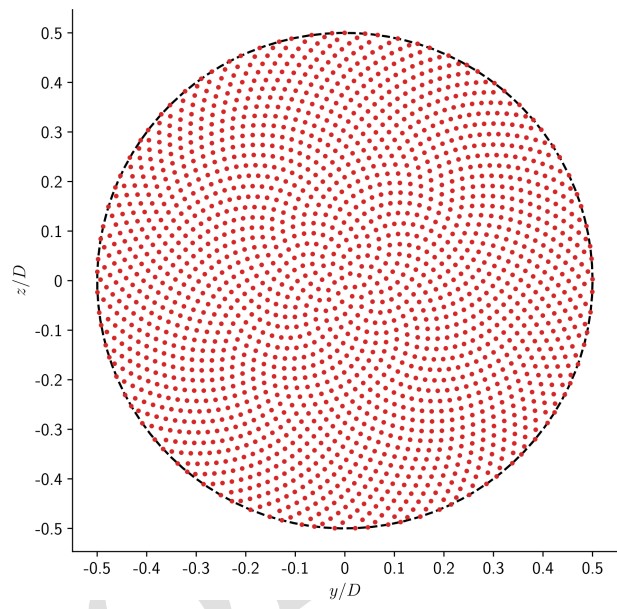

**Figure E1.** A uniform sunflower distribution of 2000 points over the surface of a circle to be used to numerically evaluate the rotor-averaged deficit due to an upstream wake in the form of Eq. (1) and to provide a reference to verify the derived analytical expressions. The polar coordinates of the shown averaging points follow Eq. (E1).

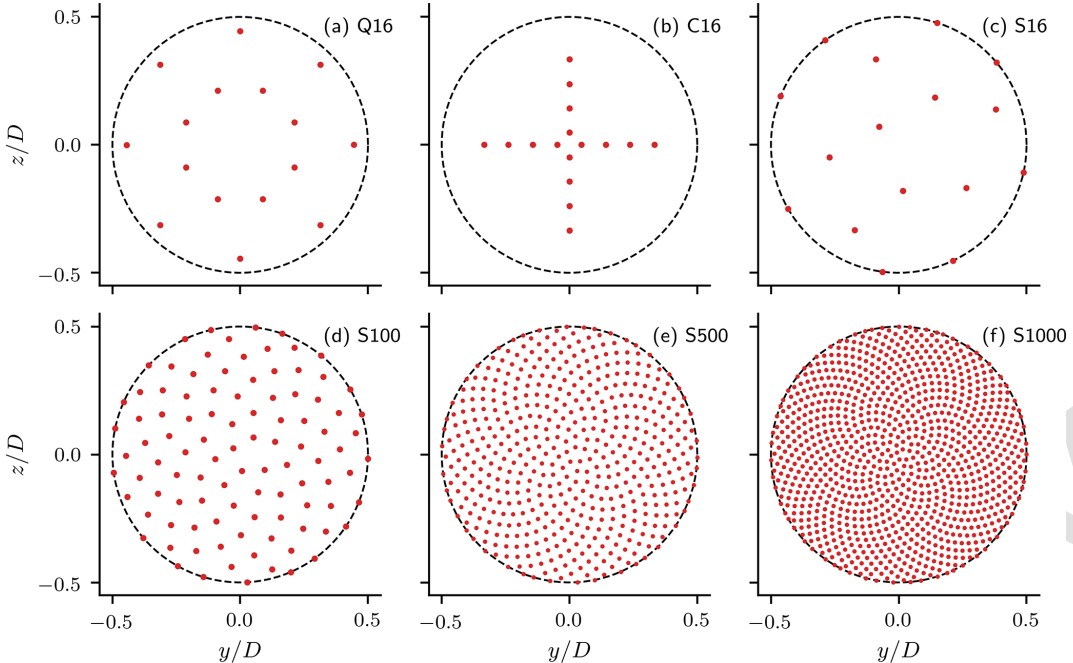

**Figure E2.** Different distributions and resolutions of averaging points following **(a)** the quadrature of Holoborodko (2011, Eq. E2), **(b)** the cross-like distribution of Stipa et al. (2024), and **(c–f)** various resolutions of the sunflower distribution (Eq. E1).

In this Appendix, we summarise various distributions and resolutions of averaging points that are used for numerical averaging of the rotor deficit.

As a reference case for verification of the derived analytical expressions (Eqs. 18 and 29) and for quantifying the uncertainty of lower-resolution numerical averaging, we use a sunflower distribution of 2000 points. For a sunflower distribution, the polar coordinates $r_k$ and $\theta_k$ of a point of index $k$ (out of $N$ total points) on a circle of radius $R$ are defined as

$$\frac{r_k}{R} = \begin{cases} 1 \text{ if } k > N - b \\ \sqrt{(2k-1)/(2N-b-1)} \text{ otherwise,} \end{cases}$$

and $\theta_k = \frac{2\pi k}{\varphi^2}$, (E1)

where $\varphi = (\sqrt{5}+1)/2$ is the golden ratio, and the constant $b = \text{round}(2\sqrt{N})$ with the function "round" returning the nearest integer. The resulting distribution of points is shown in Fig. E1. We also explore the distribution of 16 averaging points following the quadrature of Holoborodko (2011). For this quadrature, the polar coordinates of a point of index $k$ are

$$\frac{r_k}{R} = \sqrt{\frac{3 + (-1)^{k+1}\sqrt{3}}{6}},$$

and $\theta_k = \frac{2\pi(k-1)}{16}, \forall k \in \{1, 2, 3, \ldots, 16\}$. (E2)

Figure E2 shows different resolutions and distributions of averaging points including (a) the quadrature in Eq. (E2), (b)

the cross-like distribution of 16 averaging points following Stipa et al. (2024), and (c–f) various resolutions of the sunflower distribution (Eq. E1).

## Appendix F: Emphasis on the order of rotor averaging and wake superposition for a product-based superposition model

In this Appendix, we examine the effect of the order of applying wake superposition and rotor averaging for the product-based wake superposition model of Lanzilao and Meyers (2022). The following analysis is generic for any averaging order $n > 0$, but for shortness the superscript $^{(n)}$ is dropped. Consider a wind turbine impacted by a set $S$ of upstream wakes. Assuming a set of $N$ discrete points on the rotor disc of the considered turbine, the numerical approach (wake superposition followed by rotor averaging) of obtaining the rotor-averaged deficit is

$$^{\text{num}}\overline{W}_{\text{prod}} = \frac{1}{N} \sum_{k=1}^{N} \left(1 - \prod_{j \in S} \left(1 - W_j(k)\right)\right),$$ (F1)

where $W_j(k)$ is the normalised wind-speed deficit of a point of index $k$ on the rotor disc of the considered turbine due to the wake of an upstream turbine of index $j$. The product over

the set $S$ in Eq. (F1) can be expanded as

$$\prod_{j \in S} \left( 1 - W_j(k) \right) = 1 - \sum_{j \in S} W_j(k)$$
$$+ \sum_{\substack{i,j \in S \\ i \neq j}} W_i(k) W_j(k) + \mathcal{O}(W^3). \quad \text{(F2)}$$

We can neglect the higher-order terms of $W$ (order 3 and higher) compared to the lower-order terms (since $W < 1$), and hence Eq. (F1) simplifies to

$$^{\text{num}}\overline{W}_{\text{prod}} \approx \sum_{j \in S} \left( \frac{1}{N} \sum_{k=1}^{N} W_j(k) \right)$$
$$- \sum_{\substack{i,j \in S \\ i \neq j}} \left( \frac{1}{N} \sum_{k=1}^{N} W_i(k) W_j(k) \right). \quad \text{(F3)}$$

If the rotor averaging over a set of $N$ points asymptotically approaches the exact average (i.e. $N^{-1} \sum_{k=1}^{N} W_j(k) \sim \overline{W}_j$), then

$$^{\text{num}}\overline{W}_{\text{prod}} \simeq \sum_{j \in S} \overline{W}_j - \sum_{\substack{i,j \in S \\ i \neq j}} \overline{W_i W_j}. \quad \text{(F4)}$$

Alternatively, the corresponding analytical approach (rotor averaging followed by wake superposition) of obtaining the rotor-averaged deficit is

$$^{\text{anl}}\overline{W}_{\text{prod}} = 1 - \prod_{j \in S} \left( 1 - \overline{W}_j \right) \approx \sum_{j \in S} \overline{W}_j - \sum_{\substack{i,j \in S \\ i \neq j}} \overline{W}_i \overline{W}_j. \quad \text{(F5)}$$

The difference between the numerical and analytical approaches originates from $\overline{W_i W_j}$ in Eq. (F4) versus $\overline{W}_i \overline{W}_j$ in Eq. (F5), where the difference between these two quantities acts as a covariance for the set of upstream deficits. If the mutual impacts between the upstream turbines are neglected by assuming the turbines operate almost independently (i.e. $\overline{W_i W_j} \sim \overline{W}_i \overline{W}_j$), then an asymptotic resemblance between $^{\text{num}}\overline{W}_{\text{prod}}$ and $^{\text{anl}}\overline{W}_{\text{prod}}$ is achieved. In the case of small-enough deficits, this product-based superposition model approaches a non-weighted linear superposition when $W^2 \ll W$.

## Appendix G: Additional material

Here, additional material to the main text is included. We compare the rotor-averaged deficit for a circular disc (Eq. 18) with the nacelle deficit $\hat{W}$, which can be derived from Eq. (1) by substituting $\langle y', z' \rangle = \rho \langle \cos\delta, \sin\delta \rangle$:

$$\frac{\hat{W}}{C} = \exp\left( \frac{-\rho^2(\cos\delta + \omega\sin\delta)^2}{2\sigma^2(1-\xi^2)} \right) \exp\left( \frac{-\rho^2\sin^2\delta}{2\sigma^2} \right). \quad \text{(G1)}$$

Under the same conditions as in Sect. 3.1, Fig. G1 presents the offset variation of the normalised linear deficit for a circular disc ($\overline{W}_c^{(1)}$; solid) compared with the nacelle deficit ($\hat{W}$; dashed) across different values of $\delta$ at multiple downstream locations. This comparison reveals that the nacelle deficit does not adequately represent the rotor-averaged deficit, particularly at zero offset ($\rho = 0$), where $\hat{W}/C = 1$ by definition, whereas the normalised rotor-averaged deficit lies approximately between 0.6 and 0.7 for the considered cases. As such, we recommend using rotor averaging (either analytically or numerically) for applications that require a representative wind speed to estimate a turbine's operating point.

Figure G2 shows the variation of the linear rotor-averaged deficit ($n = 1$) for the circular- and rectangular-disc solutions with the offset $\rho$ at various yaw misalignments of the wake source. Both turbines have the same hub height ($\delta = 0$), and no veer effects are considered ($\Delta\alpha_o = 0$). Both the circular- and rectangular-disc solutions agree well with the numerical solution (markers) for all yaw misalignments and at all downstream distances. The impact of the yaw misalignment on the rotor-averaged deficit is small, even for $\gamma_o = 30°$, compared to wind-veer effects presented in Sect. 3.2.

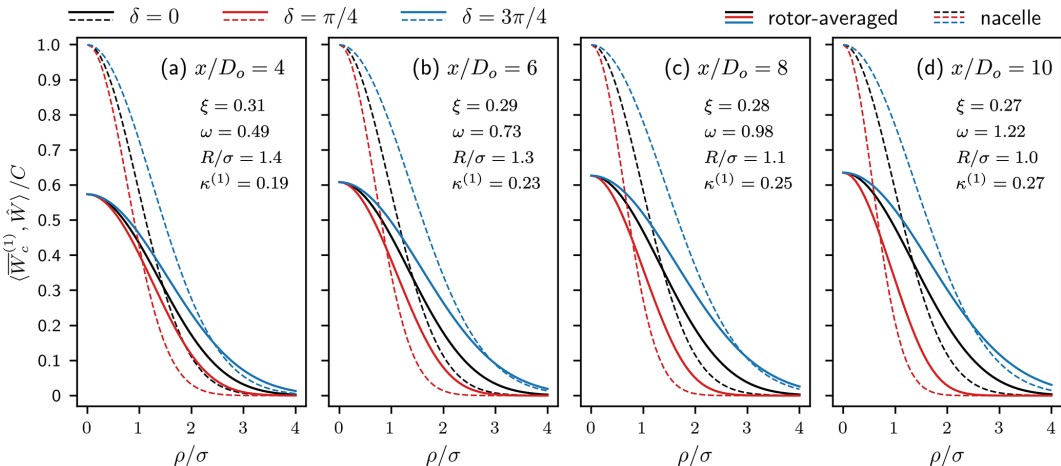

**Figure G1.** Comparing the linear rotor-averaged deficit (solid), assuming a circular disc (Eq. 18), to the nacelle wind-speed deficit (dashed; Eq. G1) for different values of the angle $\delta$. The free-stream conditions and the setting of the upstream turbine (wake source) are the same as in Sect. 3.1 and Fig. 2. The bra–ket notation in the label of the vertical axis takes the form $\langle t_1, t_2 \rangle$, which means $t_1$ or $t_2$.

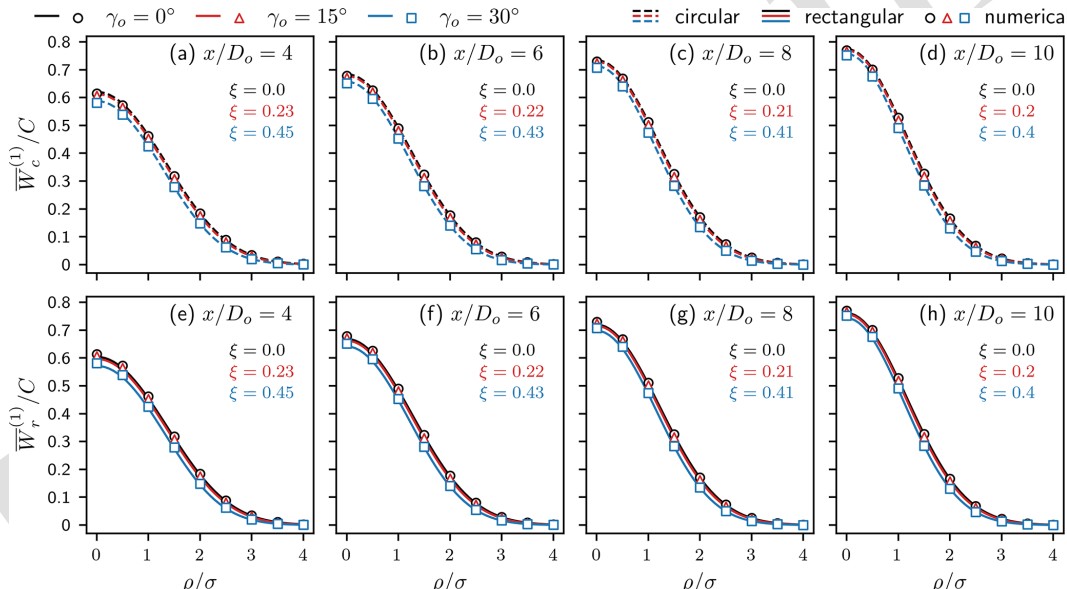

**Figure G2.** Same as in Fig. 2 but with no wind veer ($\omega = 0$) and variable yaw misalignment $\gamma_o$. The top row corresponds to the circular-disc solution (Eq. 18), and the bottom row is for the rectangular-disc solution (Eq. 29). The considered turbines have the same hub height as the wake source, and hence $\delta = 0$. The value of the eccentricity $\xi$ (Eq. 3) is indicated for each case.

**Code availability.** A Python implementation of the presented analytical expressions for the rotor-averaged deficit (Eqs. 18 and 29) is publicly available from Ali et al. (2024c) (https://doi.org/10.5281/zenodo.14622149).

**Data availability.** All data underlying this study are available in the publication or cited in the reference section. TS11

**Author contributions.** All authors contributed to the conceptualisation of this study. KA derived the mathematical expressions and prepared the original draft of the manuscript, which was reviewed and edited by TS and PO. Funding acquisition and supervision were done by TS and PO.

**Competing interests.** The contact author has declared that none of the authors has any competing interests.

**Financial support.** This research has been partly supported by the Engineering and Physical Sciences Research Council (grant no. EP/Y016297/1: Supergen Offshore Renewable Energy Impact Hub). The work has also been partly supported by the Dame Kathleen Ollerenshaw Fellowship that Pablo Ouro holds at the University of Manchester.

**Review statement.** This paper was edited by Emmanuel Branlard and reviewed by two anonymous referees.

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

## Remarks from the typesetter

**TS1** We have adjusted the size of the parentheses as requested but we have not added the times symbols or fixed the mathematical expressions on page 7 yet. Note that meaning and content changes, including changes mathematical expressions, should be reviewed by the editor before being implemented in the proofreading stage. Please reassess if these changes are strictly necessary before taking this step. For more information, please see our proofreading guidelines at: http://publications.copernicus.org/for_authors/proofreading_guidelines.html. If you want us to change the expressions on page 7 (and the times symbols), please prepare an explanatory document (doc or pdf) which we can send to the editor via our system.