# Peer review of "Direct integration of non-axisymmetric Gaussian wind-turbine wake including yaw and wind-veer effects"

_Wind Energy Science, 2024_

## Author Comment (AC1)

Dear Editor(s),

Please find enclosed the revised version of our previous submission entitled "Direct integration of non-axisymmetric Gaussian wind-turbine wake including yaw and wind-veer effects" with manuscript number WES-2024-107. We would like to thank you and the reviewers for the valuable comments which have helped to improve the quality of our manuscript. A summary of main modifications and a detailed point-by-point response to the reviewer's comments are given below. Each of the comments are typeset in blue boxes, and details of the changes that we have made to the manuscript are typeset in grey boxes.

Sincerely,

Karim Ali, Tim Stallard and Pablo Ouro

**Authors' Response to the Editor**

Following the interesting comments raised by the reviewers, we have amended the content and restructured the manuscript.

The main change in the manuscript is the introduction of a second analytical approach to obtain the rotor-averaged deficit by modelling a turbine's rotor using an equivalent rectangular disk. As now detailed in the manuscript, this approach offers several advantages over the integration over a circular disk presented in the original manuscript. In particular, this avoids dependence on complicated mathematical expressions which necessitated simplifying assumptions. The rectangular disk approach thus offers a wider range of applicability, maintains high accuracy over this range, and requires lower computational cost than the circular disk approach and when compared to approximations with numerical approaches. We updated the repository that contained a Python implementation of the circular-disk solution to also include the newly proposed rectangular-disk solution.

Further changes to the original manuscript include:

- Inspired by a comment from reviewer 1 and exploiting the Gaussian nature of the assumed wake shape, we extended the rotor averaging process from linear averaging (i.e., surface integration of W; where W is the wind-speed deficit) to rotor averaging of a generic order n > 0 (i.e., surface integration of  $W^n$ ), which is outlines in section 3.4 (updated manuscript).
- We added analysis (in sections 3.6 and 3.7; updated manuscript) addressing the computational cost of the proposed analytical expressions and their uncertainty compared to numerical averaging. Results indicated that the newly proposed rectangular representation of a turbine's rotor is faster than numerical approaches with as few as 16 averaging points whilst as accurate as using 100 averaging points.
- The validation section has been retitled verification to reflect the intent and scope, and in this section have limited the analysis to one value of thrust coefficient (0.8) and one value of turbulence intensity (5%). In the original manuscript, we had verifications of the proposed circular-disk solution at various thrust coefficients and turbulence intensities. However, these different cases only contribute to the ratio  $R/\sigma$ , where R is the turbine's radius and  $\sigma$  is the wake's standard deviation as detailed in the manuscript. Reduction of these sections has also allowed space to include verifications of different veer strengths and averaging orders as well as analysis of compute costs and uncertainty.

Importantly, we wanted to emphasise on the purpose of this study, which is not to replace numerical approaches. Such approaches are inevitable for arbitrary wind profiles. However, the Gaussian wake representation is common for typical inter-turbine spacing, and having an analytical expression for the rotor-averaged deficit can spot light on the role of key features regarding engineering wake models.

We thank you again for taking the time to consider our study for publication in Wind Energy Science. Our responses to the reviewers' comments are listed below.

**Authors' Response to Reviewer 1**

**General Comments.** The use of engineering wake models, and hence their development, is still of key interest to the wind industry. Over recent years, the individual building blocks of such models have been rightfully challenged and revised. This paper presents a rigorous exploration of one key aspect of such models that has received comparatively little attention, and in so doing derives a analytic solution to the Rotor Averaged Wind Speed problem that may significantly improve the computational speed of such models. The mathematical derivation is clear, well linked to the physics of the problem, and relies on deep mathematical insights.

**Response:**

Thank you for your feedback. We appreciate the effort put into compiling these comments. We have carefully addressed all the issues item by item as follows.

**Comment 1**

The wake model of Bastankhah and Porté-Agel [1] is used extensively to demonstrate the solution, however the solution would be applicable to a wide range of other wake models in which the Gaussian profile is used. The wider applicability of this result should be more clearly stated in section 2.1.

**Response:** Thank you for the comment.

We added the following to the preamble of section 2.

The presented analysis is applicable to any engineering wake model that utilises the Gaussian wake profile to describe the wake shape normal to the streamwise direction.

**Comment 2**

Rather than "Rotor Average Windspeed", many wake models use Root Mean Squared (RMS) speed or "Root Mean Cubed" (RMS) in calculation of thrust or power respectively. Assuming this method cannot be readily extended to RMS and RMC, a note to this restriction should be made in the text.

**Response:**

Thank you for bringing this to our attention. Actually, the presented solution can be easily extended to any averaging order n > 0 by exploiting the Gaussian form of the wake. Since  $W \propto e^{-r^2/(2\sigma^2)}$ , we have  $W^n \propto e^{-nr^2/(2\sigma^2)}$ . As such, the rotor averaged deficit across a disk of area A for an averaging order n calculated as

$$\overline{W}^{(n)} = \left(\frac{1}{A} \iint W^n \ dA\right)^{1/n},\tag{1}$$

can be obtained using the substitution  $\sigma^2 \rightarrow \sigma^2/n$ . We have updated the derivation of the rotor-averaged deficit for a circular disk to be generic for any n > 0, and introduced the rectangular-disk solution similarly (i.e., for any n > 0).

As outlined in the original manuscript, to obtain an expression for rotor-averaged deficit in the case of a circular-disk representation of the turbine, simplifying assumptions were made regarding the level of skewness of the wake contours being small. The skewness parameter ( $\kappa$ ; now denoted  $\kappa^{(n)}$ ) was used as a monitor for the level of wake skewness. Using higher-order averaging (e.g., n = 2 or n = 3) will push the circular-disk solution to the limits of its validity range by raising the skewness of the wake contours ( $\kappa^{(n)} \propto n$ ). Conversely, the newly proposed rectangular-disk solution is not limited by these simplifying assumptions as evident from the verification in Fig. 5 (updated manuscript).

The role of the averaging order is discussed in more detail in section 3.4 of the updated manuscript and emphasised in the discussion section (e.g., lines 452–461; updated manuscript).

**Comment 3**

Many wake models only use the "nacelle wind speed" (i.e. no rotor averaging) in order to reduce computational cost. I would recommend:

- 1. In the abstract "These interactions are typically quantified for each turbine by evaluating its rotor-averaged wind speed" be amended,
- 2. The nacelle point wind speed be included on the axes in figure 2 to highlight the benefit or rotor average wind speed over single point wind speed.

**Response:**

Thank you for highlighting this. The relevant sentence in the abstract was amended to

These interactions are typically quantified for each turbine either by measuring its nacelle wind speed or by evaluating its rotor-averaged wind speed using numerical methods that involve a set of discrete points across the rotor disk.

Additionally, we included the nacelle wind speed deficit in Fig. G1 of the updated manuscript to highlight the difference between this value and a rotor-averaged deficit.

**Comment 4**

The following 2 comments relate to all "rotor average wind speed" methods, but should also be considered in the text:

- 1. The impact of rotor induction perturbing the inflow profile (i.e. the rotor average speed the rotor experiences could be different from that calculated here)
- 2. The impact of the blade geometry (i.e. in a real turbine, the wind-speed at the nacelle is much less important than the wind speed at 2/3 of the blade length).

**Response:** Thank you for the comment.

We added the following to the introduction section:

Although rotor-induction effects can alter the onset wind profile of the considered turbine, we do not consider these effects similar to various engineering wake models. Additionally, we assume that the considered turbine is modelled as a uniform actuator disk, corresponding to uniform averaging weights across the turbine's rotor, and that the effects of blade geometry are neglected.

**Comment 5**

Line 23: Jensen 1983 also proposed a "Cosine-bell" profile.

**Response:** Thank you for the comment.

The sentence containing this part as amended to

including top-hat profiles (Jensen, 1983), Gaussian profiles (Bastankhah and Porté-Agel, 2014), double-Gaussian profiles (Keane et al., 2016), super-Gaussian profiles (Blondel and Cathelain, 2020), Cosine-bell profiles (Jensen, 1983; Zhang et al., 2020) ...

**Comment 6**

Figure 1: this figure is not that clear given the number of measurements that must be shown. Perhaps a set of orthographic views would be clearer?

**Response:** Thank you for the comment.

The figure has been updated with simpler 2D versions.

**Comment 7**

Line 121: "solution" (end of line) should be "approximation" or "approximate solution".

**Response:** Thank you for the comment. This sentence has been amended to

By employing Eq. 10, an approximate solution of the integral  $M_{\eta}$  in Eq. 8 is (Appendix C)

**Comment 8**

Line 136: This is the first introduction of Kappa in the text and its importance and meaning are lost. Please define kappa after Eqn. 8 or 9 (as a numbered display equation), and include a short description of it's physical meaning (i.e. "equation 8 is valid for low values of kappa. Kappa is high if...").

**Response:** Thank you for the comment.

We added a separate equation for  $\kappa$  (Eq. 14; updated manuscript) and an explanation of the physical meaning of this parameter was also included (lines 185–188; updated manuscript).

Line 247 to end of page: "the number of turbines with non-negligible deficits"... In large windfarms, the sum of a large number of upstream "negligible" wakes becomes extremely significant. It is not safe to "neglect" the large number of up-stream turbines just because each one has a small impact, as this results in significant deltas in total windfarm power.

**Response:** Thank you for the comment.

The purpose of this sentence was not to neglect smaller deficits compared to bigger ones. We were showing that the "deficit-weighted" averaged wake standard deviation  $\overline{\sigma}$  is mainly driven by 2–3 turbines with large deficits acting on the considered turbine making the exponents  $e^{-1/(4\overline{\sigma}^2)}$  (for analytical approach) and  $e^{-2/(9\overline{\sigma}^2)}$  (for numerical approach) close. This was in the context of analysing the impact of the order of applying wake superposition and rotor averaging, and is not related to calculating the overall deficit of a turbine for which all upstream wakes are considered.

To clarify this we amended this sentence to be

... where  $\overline{\sigma}$  is a deficit-weighted averaged wake standard deviation for all the upstream turbines ...

 M. Bastankhah and F. Porté-Agel, "A new analytical model for wind-turbine wakes," *Renewable Energy*, vol. 70, pp. 116–123, 2014, Special issue on aerodynamics of offshore wind energy systems and wakes. DOI: https://doi.org/10.1016/j. renene.2014.01.002.

**Authors' Response to Reviewer 2**

**General Comments.** The authors derive an analytical expression for the rotoraveraged wake velocity deficit downstream of a wind turbine, building on their previous work to now include non-axisymmetric wakes due to yaw misalignment or wind veer. The wake model predictions from this method are compared to those of the standard approach, which numerically integrates the velocity deficit at a set of discrete points across the rotor, and show excellent agreement.

The presented work is interesting and demonstrates mathematical rigor, but my main concern is demonstrating whether this model improves on the numerical averaging technique. I think the scope of this journal paper needs to be expanded to provide evidence that the work is advantageous over current methods and a convincing contribution to the literature.

**Response:**

Thank you for your feedback. We appreciate the effort put into compiling these comments.

As for the concern regarding improving over numerical techniques, we want to highlight two points. First, the purpose is not to replace numerical averaging, which is inevitable for an arbitrary wind field, but to present an alternative analytical approach for the specific case of Gaussian wakes that follow Eq. 1. Analytical approaches have an advantage over numerical ones in the sense that the role of key parameters can be elucidated.

We have now included further comparison to current methods (numerical integration) to identify differences of compute cost and uncertainty. The newly proposed rectangular-disk analytical solution is approximately 10% faster than 16-point numerical averaging, a speed-up that can benefit optimisation studies (section 3.6 in updated manuscript). Furthermore, the rectangular-disk solution has higher accuracy (less uncertainty) than the well established 16-point quadrature in the case of high wind veer (see section 3.7 in updated manuscript).

We have carefully addressed all the raised issues item by item as follows.

There seem to be two ideas that motivate the derivation of this analytical model. First, according to the introduction (line 36), "uncertainties can arise from the number, distribution, and averaging weights of the control points" used in the numerical integration process. Second, according to the discussion section (line 282), there is a use case for a differentiable wake velocity deficit model to obtain gradients for rotor-averaged wind speed with respect to turbine positions and operating parameters, which can be applied to optimization problems such as yaw control or plant layout.

Regarding the first point, I do not think this paper adequately explains how the proposed method addresses these uncertainties. Section 3 is titled "Validation," but this is really a comparison between two low-fidelity models, which I would argue is more of a verification or benchmarking process than a validation process. Without a comparison to, say, large eddy simulation results or wind tunnel experiments, how can we conclude that this model is an accurate prediction of the rotor-averaged velocity? The paper does establish good agreement between the proposed analytical method and the standard numerical approach, but it cannot make the case that it improves on the predictions of the numerical integration method.

**Response:** Thank you for the comment.

We changed the name of this section from Validation to Verification. We agree this section's purpose is just to verify the accuracy of analytical averaging compared to numerical averaging. Comparing the analytical solutions to higher fidelity data (e.g., LES) would be a validation of the Gaussian-wake assumption (Eq. 1), which is not within the scope of this study.

Following up on our discussion above, the proposed analytical solution is not intended to replace numerical averaging techniques, but to provide an alternative approach in the case of a Gaussian wake, since this wake form is widely used. Nevertheless, compute costs and accuracy are now assessed relative to numerical integration. The proposed rectangular-disk solution is approximately 10% faster than 16-point averaging (section 3.6 in updated manuscript), with less uncertainty for the case of high wind veer (section 3.7 in updated manuscript).

We included a similar discussion to the introduction section to make this clearer (lines 58–62; updated manuscript).

I think the authors could argue that this proposed analytical method agrees very well with current practices, and it is a more practical/easy to use model in design applications. One reason could be reduced computational cost, as Referee #1 states, but unfortunately the authors do not discuss how this compares between their method and the numerical approach. I suspect from the complexity of the final expressions that this cost is not trivial.

**Response:** Thank you for the comment.**

We included a dedicated section (3.6; updated manuscript) for computational cost, where 16-point averaging resolution was chosen as a reference case. The circular-disk solution was approximately 15% slower than the 16-point averaging, whereas the rectangular-disk solution was approximately 10% faster. In general, both analytical solutions have a computational cost that is comparable to vectorised calculation of the rotor average.

**Comment 3**

Regarding the second point, the authors do argue that because the model is an analytical expression of the rotor-averaged velocity, it is differentiable and can be applied to these optimization problems more easily than the numerical method. However, this paper does not derive those expressions, and I wonder how manageable the analytical formulation of that gradient would be when you consider the derivative of the proposed model, plus the wake deflection model used to obtain the wake centerline deflection, and then the expression for turbine power production. Other numerical approaches exist (i.e., algorithmic differentiation) that can calculate these derivatives for the numerical approach, so the presence of the analytical derivatives of the wake model does not automatically translate to superior optimization performance.

**Response:** Thank you for the comment.

In terms of obtaining analytical gradients, there are two approaches for this. One can differentiate the proposed solutions (Eqs. 18 and 29; updated manuscript) directly. Alternatively, one can differentiate the original surface integral (Eq. 6; updated manuscript) to obtain a set of typically similar/simpler integrals, to which the presented derivations will be a guide.

Given the range of novel topics already covered, we think it is sufficient for the current study to introduce two analytical solutions, verify them under various conditions, quantify their compute cost and uncertainty, and discuss their compatibility with wake superposition models. We certainly consider these analytical gradients for a future work.

So, I think the authors need to clearly establish the contribution of this analytical formulation. If the objective is to reduce the error of the rotor-averaging process by evaluating an analytical integral instead of a numerical integral, then I think we need to see the velocity predictions compared against proper validation data. Or, if the objective is to improve performance in an optimization application, then I'd like to see the proposed method applied to a case study where the benefit of the differentiable expression is clear, or at least a comparison of computation time between the methods.

**Response:** Thank you for the comment.

This is covered in response to comments 1–3. In summary, the analytical solutions are based on the assumption of a Gaussian wake. A comparison against higher-fidelity data would be a validation to the Gaussian-wake assumption rather than a verification of the analytical averaging. Computational-cost analysis and uncertainty quantification compared to high-resolution numerical averaging were included in the updated manuscript. The point on analytical gradients is addressed in response to an earlier comment.

**Comment 5**

Line 93: The important of accounting for wind veer is discussed in this paper, but what about shear? Both have a significant impact on power production (https://onlinelibrary.wiley.com/doi/full/10.1002/we.2917) and contribute to more complex wakes (https://iopscience.iop.org/article/10.1088/1742-6596/753/5/052004).

**Response:** Thank you for the comment.

Indeed wind shear can have an impact on the wake shape. However, the starting point in this study is the Gaussian wake shape (Eq. 1), which is common in many engineering wake models. Having an analytical solution for an arbitrary wind-speed field (in terms of shear and veer, etc) is beyond this study, if at all doable, and numerical techniques are inevitable in this case.

**Comment 6**

Line 205: Can all three of these superposition methods be defined for the reader?

**Response:** Thank you for the comment.

The three superposition models are mathematically defined in Eq. 14 (original manuscript) and in Eq. 34 (updated manuscript). Further details on each method are available in the references identified on lines 375–376.

**Comment 7**

Figures 2 and 3: A legend here indicating that the lines are the analytical model and the markers are the numerical model would help with readability.

**Response:** Thank you for the comment.

All figures in the updated manuscript have this detail.

Comment 8

Also Figure 3: I don't think the "no yaw" curves here are necessary since no comparisons between the models are done with this data and it adds clutter to the image.

**Response:**

We have updated the figure by removing the no-yaw case and grouping the three wake superposition models in one sub-figure.

**Comment 9**

Section 3.2: Can power production be defined here? It is not explicitly stated to the reader how power depends on wind speed (or yaw angle).

**Response:**

Thanks for noticing that power is not defined. In the updated manuscript, power is defined in Eq. 35.

Line 272: It is discussed here that the simplification in the derivation that the considered turbine is normal to the free-stream flow is negligible. I am wondering how the calculations for the numerical method are performed in this case—is the rotor plane taken at the yaw-misaligned angle, or is it making the same simplification as the derived analytical expression?

**Response:**

Our numerical solution is consistent with the code FLORIS, where we assume that all points on the rotor of the considered turbine have the same x coordinate (assuming that x is the streamwise direction). This makes the parameters C,  $\sigma$ ,  $\xi$ , and  $\omega$  the same for all of the points. So, the numerical solution has the same simplification as the analytical solution (lines 143–147; updated manuscript).

**Comment 11**

Line 284: Not to be nitpicky, but I think this is an oversimplification of these control/design optimization problems. The analytical formulation of the gradient of the rotor-averaged wind speed with respect to some design variables of the upstream turbine could be used in a gradient-based approach to solve the optimization problem. However, I don't think it's fair to say that it would reduce the problem to a simple root-finding problem, or that it would be able to find the global optimum.

**Response:** Thank you for the comment.

We agree this sentence was a source of confusion and over-simplification. As such, it was removed from the updated manuscript.

---

## Referee Report (RR1)

I commend the authors for the extent of their revisions and the detail and rigor of this manuscript. It is clear that the authors conducted a significant amount of analysis (and all of the figures are very sharp and easy to read), and this work will be an excellent contribution to the analytical wake modeling literature. I recommend this manuscript for publication with some minor revisions, as noted below, concerning some clarifying remarks and the summary of the contributions in Sections 4 and 5.

**Line 249:** Is this ratio L/R = 0.9 similar to the other references (DiDonato and Jarnagin, Ali, Cheung) that use this circle-rectangle analogy? It would be helpful for the reader to know if the authors' reasoning is consistent with the other similar methods in the literature.

**Line 277:** I believe the empirical expression for the wake expansion rate k* = 0.3837*TI + 0.003678 comes from this Niayifar and Porté-Agel paper ([https://www.mdpi.com/1996-1073/9/9/741](https://www.mdpi.com/1996-1073/9/9/741)) rather than the 2014 Bastankhah and Porté-Agel one referenced by the authors.

**Figure 2, Line 293:** Is there any hypothesis for why the analytical predictions for the circular disk break down for rho=0, and why the rectangular integral performs better? Also, this trend is visible for x/D = 6-10, not just at 10D downstream.

**Figures 2-3:** Would the authors consider combining Figures 2 and 3, as they have done with the data in Figure 4? I'm wondering if it would be easier to compare the rectangular and circular disks if they are plotted on the same figure. If the authors find that this presentation would be too cluttered, I think it is fine as is.

**Section 3:** There are a few points throughout this section where the authors refer to the accuracy of the analytical predictions relative to the numerical predictions (for example, "high accuracy" at line 322), and I think these claims could be supported by some quantitative information to clarify to the reader what the authors consider to be "accurate" versus "inaccurate": perhaps the maximum relative error of the rotor-averaged deficit predictions? I know that the authors do look at RMSE in Section 3.7 but it is specifically for different resolutions and distributions of averaging points.

**Section 4:** Overall, I find this section to be a bit repetitive considering the amount of detail and discussion throughout Section 3. There are some new ideas presented here: the modified definition of the coordinate frame in Eq. 37, the physical meaning of different averaging orders in line 453, the superposition of multiple wakes discussed around line 493, the partial waking discussion at line 516 (also mentioned in my next comment), and the nacelle wind speed deficit at line 531. However, much of the text that I haven't highlighted here is a summary of the results presented immediately before in Section 3. My suggestion is to expand Section 5 to include much of this summarization, which would streamline the ideas presented in Section 4 and make the paper feel less dense towards the end.

**Line 514:** I think two ideas are being combined here. First, there is the definition of the wake velocity deficit and the wind speed used for normalization: zero velocity deficit means that the local wind speed is assumed to be equal to the wind speed at the location of the upstream turbine. The authors point out that this assumption—which applies to both averaging techniques—could be invalid for wind farms with heterogenous flow fields. Second, there is the "partial waking" effect, which traditionally refers to how the wake region only partially overlaps with the rotor swept area (https://wes.copernicus.org/articles/7/433/2022/), leading to a non-uniform distribution of velocity across the rotor. The averaging process across the rotor swept area is very important in these cases of partial wake overlap. I don't fully understand the connection the authors are trying to make in this paragraph—how exactly does the wake deficit normalization factor relate to the partial waking issue?

---

## Author Response (AR2)

Dear Editor(s),

Please find enclosed the revised version of our previous submission entitled "Direct integration of non-axisymmetric Gaussian wind-turbine wake including yaw and wind-veer effects" with manuscript number WES-2024-107.

We would like to thank you and the reviewers for the valuable comments which have helped to improve the quality of our manuscript. Our response to the comments by Reviewer 1 are given below.

Sincerely,

Karim Ali, Tim Stallard and Pablo Ouro

**Authors' Response to the Editor**

Many thanks for progressing and accepting our manuscript.

We have modified the text and a couple of figures according to the recommendations of reviewer 1 (detailed below), and wherever relevant changed instances of $e^x$ to $\exp(x)$.

Figures G1 and G2 in the manuscript are correctly named. What happened previously was that the Latex engine placed the figure before the title of Appendix G. We moved the figure now below the appendix title to avoid confusion.

We have also taken this opportunity to address an oversight in the data presented in the previous version. The values tabulated in table 3 were incorrectly missing a square root and division by the number of cases, which is the same for all cases. We have updated the values accordingly. This change has no impact on the drawn conclusions.

Below are our responses to the comments by Reviewer 1.

> **General Comments.** I commend the authors for the extent of their revisions and the detail and rigor of this manuscript. It is clear that the authors conducted a significant amount of analysis (and all of the figures are very sharp and easy to read), and this work will be an excellent contribution to the analytical wake modeling literature. I recommend this manuscript for publication with some minor revisions, as noted below, concerning some clarifying remarks and the summary of the contributions in Sections 4 and 5.

**Response:**

We deeply thank you for your comments and efforts to improve the quality of our manuscript. We replied to your comments below.

> ## Comment 1
>
> Line 249: Is this ratio L/R = 0.9 similar to the other references (DiDonato and Jarnagin, Ali, Cheung) that use this circle-rectangle analogy? It would be helpful for the reader to know if the authors' reasoning is consistent with the other similar methods in the literature.

**Response:** Thank you for the comment.

- DiDonato and Jarnagin used the circle-rectangle analogy to quantify error bounds for their approximate solution of a circle, but did not use a constant value of $L/R$.
- Ali et al, used a ratio $L/R = 1$.
- Cheung chose $L/R = \sqrt{\pi}/2$ to maintain the same area between the circular and rectangular disks.

This question motivated us to look deeper into Eq. 33 (in previous version; 32 in current version), which is the source for the ratio 0.9. We found that the solution to this equation takes the form

$$\frac{L}{R} \approx \left(\frac{\sqrt{\pi}}{2}\right)^{\mathrm{erf}(2\sigma/R)}, \tag{1}$$

which can be further simplified by realising that for a typical inter-turbine spacing $2\sigma/R \gg 1$, leading to $\mathrm{erf}(2\sigma/R) \sim 1$. As such we used $L/R = \sqrt{\pi}/2$, which is approximately 0.886 i.e., very close to 0.9 which we previously used. Using $L/R = \sqrt{\pi}/2$ marginally improved the accuracy of the rectangular-disk solution.
* * *
**Comment 2**

Line 277: I believe the empirical expression for the wake expansion rate k* = 0.3837*TI +0.003678 comes from this Niayifar and Porté-Agel paper (https://www.mdpi.com/1996-1073/9/9/741) rather than the 2014 Bastankhah and Porté-Agel one referenced by the authors.

**Response:**

Thanks for noticing this. The reference has been amended accordingly.
* * *
**Comment 3**

Figure 2, Line 293: Is there any hypothesis for why the analytical predictions for the circular disk break down for rho=0, and why the rectangular integral performs better? Also, this trend is visible for x/D = 6-10, not just at 10D downstream.

**Response:** Thank you for the comment.

As discussed in lines 176–181 (previous version), to simplify the equations, the approximation $I_0\left(n\eta^2 R^2/(2\sigma_{\mathrm{ns}}^2)\right) \sim 1$ was employed. The implication of this is that the shearing and stretching of the wake contours at $\rho = 0$ were ignored, leading to an error that increases with $x/D$, as indicated in Fig. 2. We re-emphasised this in section 3.1 by including:

> The deviations between the circular-disk results and the numerical results at zero offset ($\rho = 0$ in Figs. 2c and 2d) is primarily related to the simplifying assumption in section 2.2 (Appendix C), where $I_0\left(n\eta^2 R^2/(2\sigma_{\mathrm{ns}}^2)\right) \sim 1$ was employed.

**Comment 4**

Figures 2-3: Would the authors consider combining Figures 2 and 3, as they have done with the data in Figure 4? I'm wondering if it would be easier to compare the rectangular and circular disks if they are plotted on the same figure. If the authors find that this presentation would be too cluttered, I think it is fine as is.

**Response:** Thank you for the comment.

We have updated the figure by combining Figs. 2 and 3 into one figure (Fig. 2 in updated manuscript). As such, we also combined sections 3.1 and 3.2 to be section 3.1 (updated manuscript), which considers verification for both circular and rectangular solutions.

**Comment 5**

Section 3: There are a few points throughout this section where the authors refer to the accuracy of the analytical predictions relative to the numerical predictions (for example, "high accuracy" at line 322), and I think these claims could be supported by some quantitative information to clarify to the reader what the authors consider to be "accurate" versus "inaccurate": perhaps the maximum relative error of the rotor-averaged deficit predictions? I know that the authors do look at RMSE in Section 3.7 but it is specifically for different resolutions and distributions of averaging points.

**Response:** Thank you for the comment.

Following this suggestion, we included the mean error and maximum error between the analytical and numerical solutions for the verification cases in section 3.

For the verification of the circular-disk solution:

> The mean absolute error (difference between analytical and numerical solutions) for the circular-disk solution is approximately $7.2 \times 10^{-3}$, with a maximum error of $22.5 \times 10^{-3}$ occurring in the case of zero offset ($\rho = 0$).

For the verification of the rectangular-disk solution:

> Specifically, the mean error for the rectangular-disk solution is approximately $2.7 \times 10^{-3}$, which is approximately a third of that of the circular-disk solution, with a maximum error of $7.2 \times 10^{-3}$.

For the section on veer effects, we added:

... both the circular- and rectangular-disk solutions match the numerical solutions with high accuracy with a maximum error of $8.1 \times 10^{-3}$ for the circular disk and $5.2 \times 10^{-3}$ for the rectangular disk.

Specifically, the circular disk has mean and maximum errors of $4.5 \times 10^{-2}$ and $10^{-1}$, respectively, which is 1–2 orders magnitude higher than the errors in the case of $\Delta\alpha_o = 5°$. Conversely, the rectangular-disk solution maintains higher accuracy with mean and maximum errors of $3.9 \times 10^{-3}$ and $10^{-2}$, respectively.

Of slightly less accuracy than the lower veer cases, the mean and maximum errors for the rectangular disk are $9 \times 10^{-3}$ and $1.7 \times 10^{-2}$, respectively, which are more accurate than the circular disk results at $\Delta\alpha_o = 15°$. The larger error for this case ($\Delta\alpha_o = 45°$) compared to the previous two cases is primarily due to the empirical expression for the size of the rectangular disk (Eq. 33). Higher accuracy could be achieved if the size of the rectangular disk is optimised, even though current accuracy is acceptable.

> **Comment 6**
>
> Section 4: Overall, I find this section to be a bit repetitive considering the amount of detail and discussion throughout Section 3. There are some new ideas presented here: the modified definition of the coordinate frame in Eq. 37, the physical meaning of different averaging orders in line 453, the superposition of multiple wakes discussed around line 493, the partial waking discussion at line 516 (also mentioned in my next comment), and the nacelle wind speed deficit at line 531. However, much of the text that I haven't highlighted here is a summary of the results presented immediately before in Section 3. My suggestion is to expand Section 5 to include much of this summarization, which would streamline the ideas presented in Section 4 and make the paper feel less dense towards the end.

**Response:** Thank you for the comment.

Following your suggestion, we shortened section 4 significantly by relocating the repetitive parts to the summary in section 5.

> **Comment 7**
>
> Line 514: I think two ideas are being combined here. First, there is the definition of the wake velocity deficit and the wind speed used for normalization: zero velocity deficit means that the local wind speed is assumed to be equal to the wind speed at the location of the upstream turbine. The authors point out that this assumption—which applies to both averaging techniques—could be invalid for wind farms with heterogenous flow fields. Second, there is the "partial waking" effect, which traditionally refers to how the wake region only partially overlaps with the rotor swept area (https://wes.copernicus.org/articles/7/433/2022/), leading to a nonuniform distribution of velocity across the rotor. The averaging process across the rotor swept area is very important in these cases of partial wake overlap. I don't fully understand the connection the authors are trying to make in this paragraph—how exactly does the wake deficit normalization factor relate to the partial waking issue?

**Response:** Thank you for the comment.

We agree our phrasing in this paragraph was confusing. Since the considered wake is assumed Gaussian, which is a continuous field rather than the discontinuous top-hat wake, the integration is taken across the part of the wake field occupied by the whole rotor, and is thus directly applicable to partial waking. As such, the limitation is not partial waking itself but flow heterogeneity within the farm. If the flow is highly heterogenous, the wind speed of a point of zero deficit on the turbine rotor is not necessarily the same as the wind speed of the upstream turbine. Nonetheless, all engineering wake models of this type have limitations in predicting the wake interactions with the heterogenous background flow. We omitted the part referring to partial waking, and this paragraph now is:

> Some limitations should, however, be considered. The rotor-averaging process inherently assumes that a zero-deficit point on the rotor disk has a wind speed that is equal to that of the upstream turbine (wake source), rather than the free-stream wind speed or another background wind speed. This can have profound impacts on the rotor-averaged wind speed in the case of highly heterogeneous flow within a wind farm, such as in the case of hurricanes or extremely large wind farms. In such a scenario, all numerical and analytical approaches based on engineering wake models have shortcomings as the underlying assumptions of the wake-deficit model cannot predict the interactions between the wakes and the heterogeneous background flow.